# Sulfate triple-oxygen-isotope evidence confirming oceanic oxygenation 570 million years ago

Haiyang Wang [1,2,3], Yongbo Peng [1,4] ✉, Chao Li [2,3,5] ✉, Xiaobin Cao[1,4], Meng Cheng[2,3,5] & Huiming Bao [1,4] ✉

The largest negative inorganic carbon isotope excursion in Earth's history, namely the Ediacaran Shuram Excursion (SE), closely followed by early animal radiation, has been widely interpreted as a consequence of oceanic oxidation. However, the primary nature of the signature, source of oxidants, and tempo of the event remain contested. Here, we show that carbonate-associated sulfate (CAS) from three different paleocontinents all have conspicuous negative $^{17}O$ anomalies ($\Delta'^{17}O_{CAS}$ values down to −0.53‰) during the SE. Furthermore, the $\Delta'^{17}O_{CAS}$ varies in correlation with its corresponding $\delta^{34}S_{CAS}$ and $\delta^{18}O_{CAS}$ as well as the carbonate $\delta^{13}C_{carb}$, decreasing initially followed by a recovery over the ~7-Myr SE duration. In a box-model examination, we argue for a period of sustained water-column ventilation and consequently enhanced sulfur oxidation in the SE ocean. Our findings reveal a direct involvement of mass-anomalously $^{17}O$-depleted atmospheric $O_2$ in marine sulfate formation and thus a primary global oceanic oxygenation event during the SE.

The Ediacaran Period (635-539 Ma) witnessed the largest negative carbonate carbon isotope ($\delta^{13}C_{carb}$) excursion in Earth's history, known as the Shuram Excursion (SE, or Wonoka/DOUNCE/EN3) between 574.0 ± 4.7 and 567.3 ± 3.0 Ma[1–4]. Typically, the SE is characterized by a dramatic drop in the $\delta^{13}C_{carb}$ from as high as +5‰ to as low as −12‰ globally. The difficulty in explaining the large shift in the global carbon cycle and the often-positive correlation between $\delta^{13}C_{carb}$ and $\delta^{18}O_{carb}$ have led some researchers to argue that the SE is a product of later diagenesis (e.g., meteoric alteration[5,6] or burial diagenesis[7]). But diagenesis is inherently a local process and seems inconsistent with the global occurrence of the SE[1,3,4]. Supportive evidence for the primary nature of the SE comes from petrography[8], organic geochemistry[9], and high-spatial-resolution isotope analysis[10,11]. Consequently, the SE is interpreted as a result of enhanced oxidation of $^{13}C$-depleted organic carbon reservoirs, such as the dissolved organic carbon (DOC) in

seawater[12–14], fossil organic matter exposed on land[15], hydrocarbon-rich fluids from the subsurface[9], and/or methane emitted from the sediments[16]. The onset of the SE is closely followed by the first appearance of large, unambiguous metazoan fossils[17]. Thus, constraining the origin of the SE can help elucidate the interaction between Earth system oxygenation and the radiation of early animals.

A sustained oxidation of organics over ~7 million years (Myrs)[2,18] during the SE requires a consistent supply of oxidant, if it is indeed an oceanic oxygenation event. Multiple lines of evidence, including S-U-Tl isotope compositions of carbonate or shale[13–15,19–21] and iron speciation in shale[22], have supported an increased oceanic ventilation, i.e., rising marine concentrations of dissolved $O_2$ and/or sulfate during the SE, although a numerical model has argued that seawater oxidants would not be sufficient in driving a global carbon perturbation for millions of years[23]. The dissolution of older continental evaporite may provide

[1]International Center for Isotope Effects Research, Nanjing University, Nanjing, China. [2]State Key Laboratory of Oil and Gas Reservoir Geology and Exploitation & Institute of Sedimentary Geology, Chengdu University of Technology, Chengdu, China. [3]International Center for Sedimentary Geochemistry and Biogeochemistry Research, Chengdu University of Technology, Chengdu, China. [4]Frontiers Science Center for Critical Earth Material Cycling and State Key Laboratory for Mineral Deposits Research, School of Earth Sciences and Engineering, Nanjing University, Nanjing, China. [5]Key Laboratory of Deep-time Geography and Environment Reconstruction and Applications of Ministry of Natural Resources, Chengdu University of Technology, Chengdu, China. ✉e-mail: ybpeng@nju.edu.cn; chaoli@cdut.edu.cn; bao@nju.edu.cn

additional oxidant[24], and/or the oxidation of DOC could be spatially heterogeneous due to local oxygenic photosynthesis[25]. Nevertheless, the oxidants available for organic matter remineralization during the SE are, (1) dissolved $O_2$ in the ocean supplied from atmosphere and/or local photosynthesis, (2) newly formed sulfate from sulfide/sulfur oxidation on the continents and/or in ocean basins, and/or (3) weathered sulfate from the dissolution of older evaporite deposits on the continents. Regardless of whether $O_2$ played a direct or indirect role in oxidizing organics, the role of sulfate as an oxidant of organics could be significantly enhanced due to a rising $O_2$ concentration. This may occur through enhanced oxidative weathering of sulfide minerals on the continents[26,27] or ventilation of deeper sulfidic seawater. Therefore, the origin of the sulfate is key for understanding the nature of the SE as an oceanic oxygenation event.

Sulfur and triple oxygen isotope compositions (i.e., $\delta^{34}S$, $\delta^{18}O$, and $\delta^{17}O$), especially $^{17}O$ anomalies [i.e., $\Delta'^{17}O \equiv (\ln\frac{\delta^{17}O+1000}{1000} - 0.5305 \times \ln\frac{\delta^{18}O+1000}{1000}) \times 1000‰$], provide a powerful tool to disentangle the origin of sulfate[28]. If atmospheric $O_2$ was indeed the main oxidant responsible for the SE, an increase in sulfate concentration during the SE would occur through enhanced sulfide/sulfur oxidation as a result of a deep-water invasion of atmospheric $O_2$ or an expansion of ventilated ocean volume. Importantly, the increased sulfate would be accompanied by a negative shift in the sulfate $\Delta'^{17}O$ value of the ocean because newly-formed sulfate would inherit the atmospheric $O_2$ signature that bears a unique mass-independent $^{17}O$ depletion[28-35]. We note that such a negative shift in sulfate $\Delta'^{17}O$ value is unlikely to be generated by diagenesis, as no diagenetic process has been found to be capable of bringing more atmospheric $O_2$ into the sediments or rocks. In contrast, if the dominant oxidant is evaporite sulfate dissolved from the continents[24], it would be highly unlikely to detect distinctly negative $\Delta'^{17}O$ values for sulfate extracted from the SE carbonates because most of the pre-Ediacaran evaporites do not exhibit large $^{17}O$ depletion[31,32]. Sulfate generated from sulfide/sulfur oxidation by photosynthetic $O_2$ in the water column at local photic zones would also not have significantly negative $^{17}O$ anomalies[36]. Moreover, further insights into the trigger and dynamics of the SE oxygenation event can be obtained when the temporal trends and spatial heterogeneity of sulfate's sulfur and triple oxygen isotope compositions are combined with associated carbon isotope trends.

To test the hypotheses above, we selected SE-containing units from three paleogeographically different continents[1,3]: the Doushantuo Formation of South China, the Wonoka Formation of South Australia, and the Shuiquan Formation of Tarim, all of which have well-constrained carbon isotope records (Fig. 1; Supplementary Discussion), and we extracted carbonate-associated sulfate (CAS) and analyzed the NaCl-leached and the HCl-leached fractions for their respective $\delta^{34}S$, $\delta^{18}O$, and $\Delta'^{17}O$ values (see Methods).

## Results and discussion
### Sulfur and triple oxygen isotope data
The $\Delta'^{17}O$ values of the HCl-leached CAS range from −0.53 to −0.14‰, −0.51 to −0.07‰, and −0.35 to +0.06‰ in South China, South Australia, and Tarim, respectively (see Supplementary Data 1). The $\Delta'^{17}O$ nadirs are distinctly lower than −0.07 ± 0.09 ‰ (1σ), the average value for modern and Phanerozoic sulfate minerals (also see compiled data in refs. 31,37). Moreover, the $\Delta'^{17}O$ records in all three paleocontinents examined display a remarkably similar stratigraphic trend, with an initial decrease followed by a recovery over the SE intervals (Fig. 1). This trend is also observed for their corresponding $\delta^{34}S$, $\delta^{18}O$, and $\delta^{13}C_{carb}$ records. The $\delta^{34}S$ values of the HCl-leached CAS range from 16.3 to 42.0‰, 15.2 to 29.1‰, and 10.2 to 18.4‰ and their corresponding $\delta^{18}O$ values from 8.0 to 23.1‰, 8.8 to 15.3‰, and 10.4 to 15.0‰ in South China, South Australia, and Tarim, respectively.

### Sulfate $^{17}O$ anomaly in geological records for tracking paleoatmospheric $O_2$
The $^{17}O$ anomaly signals found in geological sulfate minerals are a powerful tool to investigate biogeochemical cycles of carbon, sulfur and oxygen over Earth's history, especially with regard to processes related directly to atmospheric $O_2$[28-33]. Atmospheric $O_2$ is the only known source compound that bears a negative $\Delta'^{17}O$ value, which originates from mass-independent fractionation during photochemical reactions involving $O_2$, $O_3$, and $CO_2$ in the stratosphere[34,35]. The $\Delta'^{17}O$ value of atmospheric $O_2$ is primarily determined by $pO_2/pCO_2$ ratios and the rates of gross primary production[34,38]. Lower $pO_2/pCO_2$ ratios or lower gross primary production would lead to more $^{17}O$-depleted atmospheric $O_2$[38]. The $^{17}O$ anomaly in atmospheric $O_2$ could be passed onto newly formed sulfate through sulfide/sulfur oxidation, as demonstrated by laboratory experiments in which approximately 8 to 30% of the oxygen atoms in the produced sulfate originated from $O_2$[39,40]. However, the $^{17}O$ anomaly formed in the sulfate can potentially be erased by sulfur redox processes, as these processes enable sulfate to exchange oxygen atoms with the surrounding water through a backward exchange between sulfite and sulfate[41] and/or via anaerobic oxidation of $H_2S/S^0$ by nitrate or Fe(III) in water or sediments[42]. Nevertheless, the preserved $^{17}O$ anomaly signals in geological sulfate minerals establish a direct association between the isotopic composition of sedimentary sulfate and paleoatmospheric $O_2$. Significantly negative $\Delta'^{17}O$ values in geological sulfate provide strong evidence for the partial incorporation of paleoatmospheric $O_2$ into sulfate, usually through sulfide/sulfur oxidation[28-33].

### Evaluating the extracted CAS
Extracted CAS from carbonate outcrops could be contaminated by present-day atmospheric sulfate[43] and/or newly generated sulfate from pyrite oxidation during outcrop weathering or laboratory experiments[44,45]. Prior to the HCl-solution extraction, repeated NaCl-solution leaching can effectively remove most, if not all, of the syn-sedimentary evaporites (if present), sulfate generated by post-depositional processes (e.g., diagenesis, laboratory treatment), and modern atmospheric deposition. Often, sulfate produced via the oxidation of sulfide minerals has lower $\delta^{34}S$ and $\delta^{18}O$ values than the original seawater sulfate[45,46].

To evaluate the degree of contamination of the non-original sulfate in the HCl-leached sulfate, we compared the sulfur and oxygen isotope compositions of the NaCl-leached and the HCl-leached sulfate. The $\Delta'^{17}O$ values of the HCl-leached sulfate are more negative (i.e., more distinct) than those of the corresponding NaCl-leached sulfate, while both the $\delta^{34}S$ and $\delta^{18}O$ values of the HCl-leached sulfate are generally higher than those of the NaCl-leached ones (Fig. 2, Supplementary Figs. 1, 2). This pattern holds true for samples from all three paleocontinents. Meanwhile, the stratigraphic trends of these S- and O-isotopes are drastically different between the HCl-leached and the NaCl-leached sulfate (Supplementary Fig. 2). These observations indicate that the NaCl-leached fraction contains a significant amount of sulfate originating from post-depositional oxidation of sulfide minerals due to their low $\delta^{34}S$ and low $\delta^{18}O$ values, and/or of present-day atmospheric sulfate due to the positive $\Delta'^{17}O$ values (up to +0.51‰)[35,47,48]. In contrast, the presence of non-original sulfate in the HCl-leached CAS is minimal. It is notable that the Tarim CAS may still have small fractions of the present-day atmospheric sulfate, as indicated by their slightly positive $\Delta'^{17}O$ values (up to +0.06‰). Thus, the $\Delta'^{17}O$ values of the original sulfate during the SE in Tarim should be more negative than the data shown here. Nevertheless, the magnitudes of the $\Delta'^{17}O$ negative shift of −0.39‰, −0.44‰, and −0.41‰ in the HCl-leached CAS during the SE in South China, South Australia, and Tarim, respectively, are almost identical, strongly favoring the observed sulfate $\Delta'^{17}O$ negative shifts being a credible primary and global signature (Fig. 1).

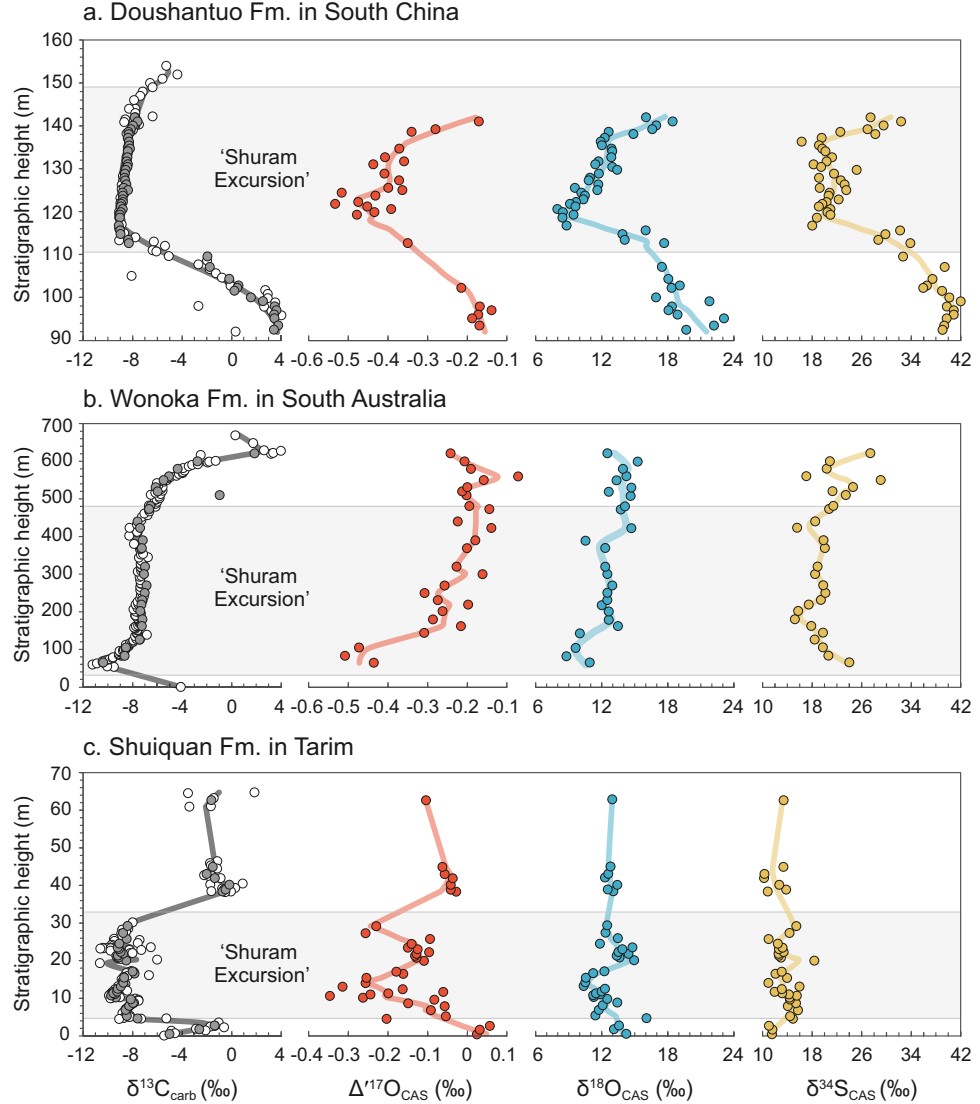

**Fig. 1 | C-S-O isotope compositions of the Shuram Excursion. a** South China, **b** South Australia, and **c** Tarim. The light-grey shaded areas refer to intervals of the largest negative carbon isotope excursion (defined here by $\delta^{13}C_{carb}$ lower than −6‰). The solid lines represent LOWESS curves. Analytical uncertainty (1σ) for isotope compositions is smaller than the sizes of the symbols. Carbon isotope data in grey-filled cycles represent the samples used in this study, while those in open cycles are from refs. 25,69,70.

Depending on the formation condition, the original sulfate residing as HCl-leached CAS can be entirely of seawater sulfate origin if the carbonate rocks were precipitated from seawater and had a negligible contribution from carbonate cements formed during early diagenesis. The SE carbonates were most likely precipitated in the water column or at the sediment–water interface, as independently supported by their calcium and magnesium isotope composition[49,50]. Furthermore, the SE is recorded in a variety of depositional facies, ranging from shallow water peritidal to open deep slope settings[1,3]. Therefore, the HCl-leached CAS we measured primarily reflects seawater sulfate of the SE if later diagenetic alteration is excluded (see discussion followed).

### Evaluation of diagenetic alteration

In addition to the near-identical sulfate $\Delta'^{17}O$ negative shift in the three separate paleocontinents, our data display tight stratigraphic co-variations of their C–S–O isotope composition (Fig. 1), supporting an original, coupled perturbation to marine carbon and sulfur cycling during the SE. Indeed, no diagenetic processes have been found to

yield significantly negative $\Delta'^{17}O$ values in sulfate, a signal unique to sulfate that carries atmospheric $O_2$ signature of geological times[28–32].

Hypothetically, $O_2$-rich meteoric water circulating through pyrite-rich carbonates may oxidize the pyrite to form sulfate with potentially negative $\Delta'^{17}O$ values if the $O_2$ had a negative $\Delta'^{17}O$ value. Such a scenario would result in the redistribution of sulfate in these carbonates and a homogenized CAS multi-isotope range in the stratigraphic level, which is not observed. In fact, a consistent C–S–O isotope variation is observed within tens of meters of carbonate formations and across three separate paleocontinents (Fig. 1). In addition, if the CAS originated from pyrite oxidation, the $\delta^{34}S$ value of the CAS would be close to that of co-existing pyrite in carbonates, and its $\delta^{18}O$ value would be much lower, often close to that of the solution water[51]. However, neither scenario is observed[13,14,20,52]. For example, the average $\delta^{34}S$ value of the HCl-leached CAS from South China is -22.8‰, substantially higher than the mean value of −7.4‰ in the co-existing pyrite[20]. The $\delta^{18}O$ value of the meteoric water is most likely lower than 0‰[53], but the measured CAS exhibits significantly higher $\delta^{18}O$ values, averaging 12.2–12.4‰ (Fig. 2, Supplementary Table 1).

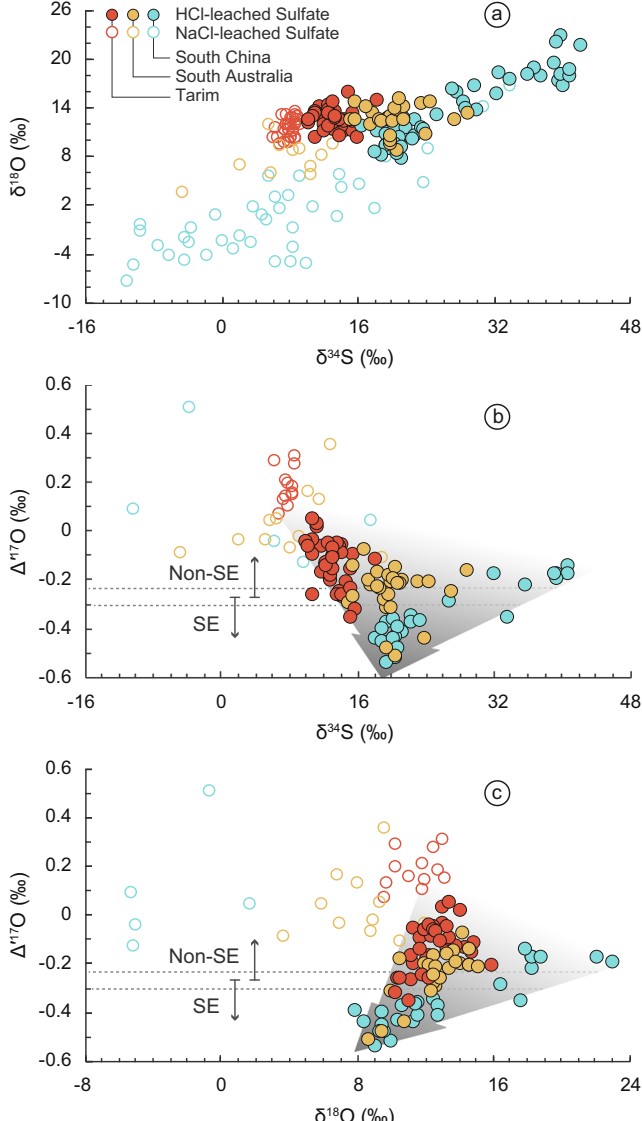

**Fig. 2 | Crossplots of $\Delta'^{17}O$, $\delta^{18}O$ and $\delta^{34}S$. a** $\delta^{18}O$ vs. $\delta^{34}S$. **b** $\Delta'^{17}O$ vs. $\delta^{34}S$. **c** $\Delta'^{17}O$ vs. $\delta^{18}O$. Arrows filled with a gradual color denote the ranges of $\delta^{34}S$ and $\delta^{18}O$ of the HCl-leached sulfate converge to a point when the $\Delta'^{17}O$ values are the most negative.

Some researchers argue in favor of a diagenetic origin of the SE, primarily based on the co-variation between $\delta^{13}C_{carb}$ and $\delta^{18}O_{carb}$ in carbonates[5,7], which is commonly observed in typical SE sections (although not universally[1,3]). While late meteoric diagenesis can generate a positive correlation between $\delta^{13}C_{carb}$ and $\delta^{18}O_{carb}$, the existence of this correlation does not necessarily confirm meteoric diagenesis. The $\delta^{18}O_{carb}$ is sensitive to changes in ambient temperature, the $\delta^{18}O$ of solution water, or late diagenesis[4,11]. Often, the $\delta^{18}O_{carb}$ is more susceptible to resetting than the corresponding $\delta^{13}C_{carb}$ due to the greater abundance of external oxygen compared to carbon during later fluid-rock interactions[3]. Therefore, the $\delta^{13}C_{carb}$ could be of primary origin, as supported by our data, even if the $\delta^{18}O_{carb}$ is influenced by diagenesis. Furthermore, recent studies, based on in-situ carbon isotope analysis[10,11], Ca and Mg isotopes and carbonate-associated phosphate concentrations over different SE sections[49,50,54], and other geochemical and stratigraphic observations[1,8,9], have argued against meteoric or late burial diagenesis as the cause of the SE. Our results cannot rule out the possibility of a diagenetic origin for the $\delta^{18}O_{carb}$ in the SE carbonates,

thus the discussion will primarily focus on sulfate multi-isotopes and their relationships with the $\delta^{13}C_{carb}$ values.

## Marine sulfate evolution

The spatial heterogeneity of sulfur and oxygen isotope compositions of the pre- and post-SE sulfate supports a scenario of low and fluctuating marine sulfate concentration. The $\delta^{34}S$ and $\delta^{18}O$ values of the pre-SE HCl-leached CAS from different paleocontinents vary widely, with average values ranging from 12.3 to 38.8‰ and from 14.2 to 19.3‰ (the South Australia section lacks a pre-SE record), respectively, whereas those during the SE cluster around 13.8–22.8‰ and 12.2–12.4‰, respectively (Fig. 2; Supplementary Table 1). These patterns are consistent with a low marine sulfate concentration preceding the SE when the sulfur and oxygen isotope compositions were susceptible to local perturbations, while during the SE an enhanced sulfate flux was supplied to continental shelves. The added sulfate appears to have the same origin worldwide and was overwhelming in quantity because its sulfur and oxygen isotope compositions are converging to the same set of values across three different continental shelves (Fig. 2).

What then, is the source of the enhanced sulfate flux? Our CAS data from three examined paleocontinents all show distinctly negative $\Delta'^{17}O$ values and similar $\delta^{34}S$ and $\delta^{18}O$ values during the SE. This observation argues against evaporite dissolution being the main source of the added sulfate input, as contribution from evaporite dissolution should be local and it is unlikely that evaporites with the same multi-isotope composition would exist on different continents. The cross-plots of $\Delta'^{17}O$ and $\delta^{18}O$ show that the pre-Ediacaran evaporites do not fall on the mixing lines and are therefore unlikely a component of the SE sulfate (see Supplementary Fig. 3). In addition, pre-Ediacaran evaporites are found in small scales, with the current volume of ~4.7 × 10⁵ km³ in total[55]. If being all gypsum and all dissolved, they could only supply ~0.6 Myr of the sulfate flux (i.e., ~1 × 10¹³ mol yr⁻¹) needed in Shields et al's model estimate[24]. Even considering dissolution of pre-Ediacaran evaporites after deposition, these numbers require an unreasonably large volume of evaporites exposed during pre-Ediacaran time, and no evidence for massive pre-Ediacaran evaporite dissolution was found.

Instead, there must be an enhanced influx of sulfate derived from sulfide/sulfur oxidation via atmospheric $O_2$ during the SE. However, the sulfate source could come from either pyrite weathering on the continents[15,25] or oxidation of sulfur compounds (e.g., $H_2S$ and $S^0$) in the deeper ocean. While the latter is less known, $H_2S$ and $S^0$ can be oxidized directly by $O_2$[56,57], and this process may occur in natural environments where both $H_2S$ and $O_2$ are present[58–60]. The two cases depict different scenarios of the SE oxygenation event. The former may suggest that the threshold of atmospheric $pO_2$ (i.e., ~0.4% of the present-day[27]) for the sensitivity of oxidative pyrite weathering had not been reached in the Ediacaran, or there was an increase in the exposed surface area of sulfide-rich sediments on the continents during the SE[15]. The latter requires the ventilation of a deeper, sulfide-rich marine water body and ocean itself being a source of sulfate via sulfur oxidation. To determine the plausibility of each hypothesis, we resort to quantitative modeling.

## Marine S-cycle modeling and the origin of the ¹⁷O-depleted sulfate

The continued stratigraphic sulfate S–O isotope records, along with the duration constrained by high-quality geochronological ages[2,18], provide an unparalleled opportunity to place quantitative constraints on sulfur geochemical dynamics. Here we used a non-steady-state box model of global sulfur cycle to estimate the source and flux of the enhanced sulfate during the SE (see Supplementary Discussion for a full description of the model and Table 2 for all parameters used). Whereas such a model has been well developed for exploring sulfur cycling in ancient oceans[61,62], this is the first time that the $\Delta'^{17}O$

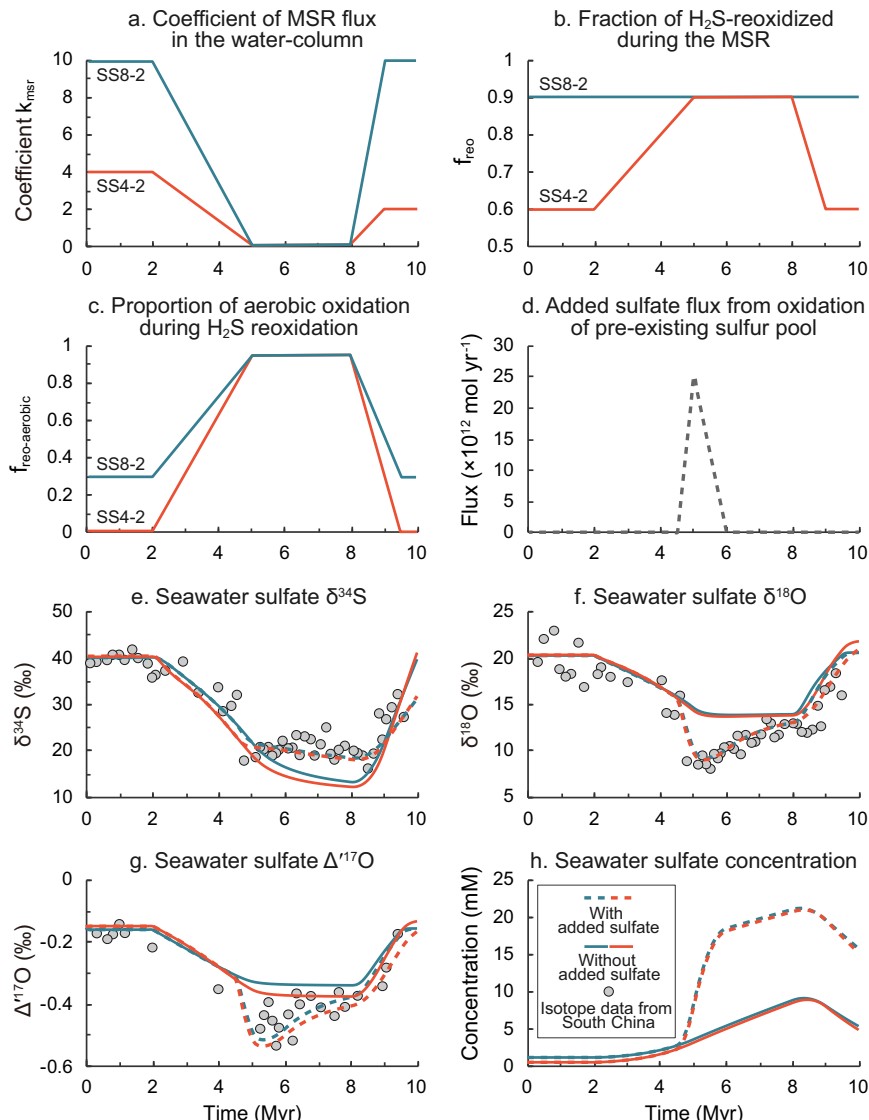

**Fig. 3 | Output of our S-cycling box-model.** Driven forces: changes in **a** coefficient $k_{msr}$ of MSR (microbial sulfate reduction) flux in the water-column, **b** fraction of $H_2S$-reoxidized during the MSR, **c** proportion of aerobic oxidation during $H_2S$ reoxidation, and **d** added sulfate flux from oxidation of the pre-existing sulfur pool. Model results: **e** seawater sulfate $\delta^{34}S$, **f** seawater sulfate $\delta^{18}O$, **g** seawater sulfate $\Delta'^{17}O$, and **h** seawater sulfate concentration. The initial steady states with sulfate concentration of 0.5 mM (SS4-2, see Supplementary Table 3) and 1.1 mM (SS8-2) were adopted as suggested[72,73].

parameter has been incorporated into the model, which constrains processes such as $H_2S/S^0$ oxidation specifically.

Assuming that all of the increased sulfate was sourced from pyrite weathering on the continents, we adopted initial steady states (see Supplementary Table 3) established based on the pre-SE's data of South China ($\delta^{34}S \sim 40‰$, $\delta^{18}O \sim 20‰$, and $\Delta'^{17}O \sim -0.17‰$) to drive changes and fit the observed temporal isotope trends. We selected South China data as a typical case for modeling because of its complete and continuous stratigraphic records, as well as its likely limited contamination to its S- and O-isotope compositions when compared to the other two sections. Our modeling results show that a rise in pyrite weathering flux ($F_{wp}$) alone cannot simultaneously reproduce the observed shifts in sulfur and oxygen isotope compositions ($\delta^{34}S \sim 20‰$, $\delta^{18}O \sim 12‰$, and $\Delta'^{17}O \sim -0.5‰$) during the SE (Supplementary Fig. 4a). The magnitude of the $\delta^{34}S$ shift is much larger than that of the $\delta^{18}O$, which is likely due to (1) a buffering effect from the intense oxygen isotope exchange between intermediate sulfur species and the ambient $H_2O$ during microbial sulfate reduction (MSR) in seawater[41,42], and (2) the significantly greater difference in $\delta^{34}S$ ($-17‰$

vs. 40‰) than that of the $\delta^{18}O$ (0‰ vs. 20‰) between the pyrite-derived and the pre-existing sulfate.

If we were to reproduce the full $\delta^{18}O$ shift using only pyrite weathering, it would require an unrealistic increase of two orders of magnitude in its flux, as well as an additional condition that the $\delta^{34}S$ of the pyrite-derived sulfate needs to be around 18‰ (Supplementary Fig. 4b), which is unreasonably higher than the sulfate $\delta^{34}S$ (close to 4.8‰[63]) in modern global riverine systems. Furthermore, if this process were to last for $\sim 7$ Myrs, it would result in an unrealistic increase of more than two orders of magnitude in marine sulfate concentration (reaching up to $\sim 300$ mM; Supplementary Fig. 4b), which is much higher than the modern value of 28 mM[42]. Therefore, we conclude that a flux increase in the $^{17}O$-anomalous riverine sulfate alone cannot reasonably explain the temporal sulfate isotope trends during the SE.

Alternatively, an increased oxidation of reduced sulfur species in seawater linked to water-column ventilation may play a vital role in controlling the observed temporal multi-isotope trends. On one hand, water-column ventilation could increase the fraction of extra-cellular aerobic $H_2S$ reoxidation (i.e., higher $f_{reo}$ and $f_{reo\text{-}aerobic}$)

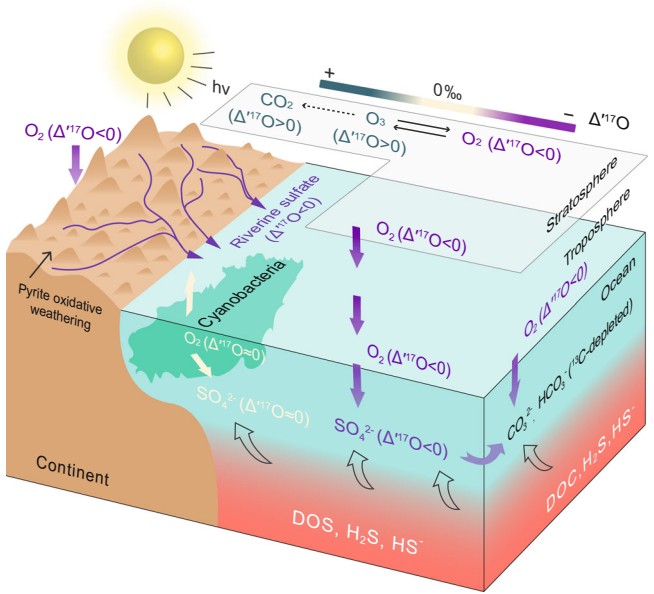

**Fig. 4 | Schematic presentation of the origin of the ¹⁷O-depleted sulfate during the Shuram Excursion.** Mass-independently ¹⁷O-depleted $O_2$, generated during photochemical reactions of $O_2$, $O_3$ and $CO_2$ in the stratosphere, mixes into the ocean through gas exchange between stratosphere and troposphere, and between atmosphere and ocean. The newly formed sulfate inherits the ¹⁷O-depleted signature from atmospheric $O_2$ through oxidation of the reduced sulfur species in the ocean and on land. See text for more details. DOC dissolved organic carbon, DOS dissolved organic sulfur.

during the course of MSR[64]; on the other, as the concentration of dissolved $O_2$ increases, it would become the dominant oxidant in organic matter remineralization, suppressing MSR in the water column (i.e., lower $k_{msr}$)[65]. These two processes both contribute to the decreases in sulfate sulfur and oxygen isotope compositions (Supplementary Fig. 4c–e). However, even when using parameters representing a well-oxygenated condition, i.e., a near-zero MSR flux in the water column ($k_{msr} = 0$; Fig. 3a) and a high fraction of aerobic $H_2S$ reoxidation ($f_{reo} = 0.9$; $f_{reo\text{-}aerobic} = 0.95$; Fig. 3b, c), the model still cannot reproduce the observed decreases in $\delta^{18}O$ and $\Delta'^{17}O$ during the SE (Fig. 3f, g), suggesting that other factors must be at play.

Sulfidic water was expected in the shelf areas preceding the SE[66], and this condition could contribute to organic carbon accumulation and provide a stable dissolved organic sulfur reservoir in the ocean through DOC sulfurization[67]. An enhanced oxidation of these pre-existing sulfur species (i.e., $H_2S$ and $S^0$) could provide an additional sulfate source with lower $\delta^{18}O$ and $\Delta'^{17}O$ values (Fig. 4). When we also factor in the oxidation of the pre-existing sulfur species, the model is capable of reproducing all of the observed S–O isotope compositions and their temporal changes (Fig. 3d-g; Supplementary Figs. 5-7). The optimum condition requires the $\delta^{34}S$ of the pre-existing $H_2S/S^0$ reservoir to be in the range of 16–25‰ [note that the pre-SE pyrite $\delta^{34}S$ averages $18.7 \pm 8.7$ ‰ (1σ) in South China[14], and the $\delta^{18}O$ and $\Delta'^{17}O$ values of its derived sulfate to be at 0–4‰ and −0.7 to −0.8‰, respectively. Additionally, the added sulfate flux from the oxidation of the pre-existing sulfur species needs to occur as a pulse of $1$–$2.5 \times 10^{13}$ mol yr⁻¹, with a duration of 1.5 Myrs (Fig. 3d). This scenario corresponds to an increase in the marine sulfate concentration from ~0.5–1 mM to ~20 mM (Fig. 3h).

The sulfate ¹⁷O-depletion during the SE occurred ~60 Myrs after the basal Ediacaran ¹⁷O-depletion event[2,68]. This raises the question of if atmosphere $O_2$ was consistently depleted in ¹⁷O throughout the Ediacaran Period and the high sulfate concentrations at the aftermath of

Marinoan Snowball Earth and during the SE merely facilitated the preservation of the sulfate ¹⁷O record, or if atmospheric $O_2$ was distinctly depleted in ¹⁷O only during these two episodes. A wider geographic coverage of a similar dataset is needed to further answer this question. Additionally, further research efforts are required to validate if sulfate derived from $H_2S$ oxidation in natural water bodies indeed incorporates the $\Delta'^{17}O$ signal of atmospheric $O_2$.

## Conclusions and implications for the SE event

Taken together, our sulfate S-O isotope data and modeling results support water-column ventilation during the SE, which led to enhanced sulfur oxidation by mass-anomalously ¹⁷O-depleted $O_2$ dissolved in the oceans (Fig. 4). The positive co-variations of the C–S–O isotope compositions observed in three separate paleocontinents imply that the processes discussed above also apply to the carbon isotope records. It could be either an increase in atmospheric $O_2$ concentration or increased ventilation of the deep oceans that facilitated the oxidation of marine organics, thus contributing to the negative carbon isotope excursion. In summary, our findings confirm the SE as a primary oceanic oxygenation event, rather than of a late diagenetic origin, and provide direct evidence for paleoatmospheric $O_2$ being ultimately responsible for the oxidation of reduced sulfur and organics in the oceans. This resolves the long-standing debate on the origin of the largest negative C-isotope excursion in Earth's history and establishes a link between environmental oxygenation and the rise of early animals.

## Methods
### Sampling
We targeted the Ediacaran carbonates from three different paleocontinents which have recorded the Shuram Excursion (Fig. 1; Supplementary Discussion). A total of 117 samples were collected, including 51 samples from the Jiulongwan section (30°47′51″N, 110°59′32″E) in South China, 28 samples from the Parachilna Gorge section (31°9′51.6″S, 138°31′43.2″E) in South Australia, and 38 samples from the Mochia-Khutuk section (41°26′29″N, 87°51′47″E) in Tarim. Samples were cut, cleaned, and ground to ca. 200 mesh for bulk geochemical analyses. Stratigraphic details for these sections can be found in refs. 14,20,69,70.

### CAS extraction and purification
Approximately 50 g sample powder was immersed in a 10% NaCl solution at least 4 times, with each immersion lasting at least 12-h. This process yielded NaCl-leached sulfate. After the final NaCl-leaching, a saturated $BaCl_2$ solution was added to the filtered and acidified solution to check if there is $BaSO_4$ precipitating. If precipitates are visible, one more NaCl leaching was conducted. Next, 4 M HCl solution was added to the NaCl-leached sample residue and let it sit in room temperature for less than 1 hour, resulting in HCl-leached sulfate. A fraction of the $BaSO_4$ precipitate was purified using the DDARP method (DTPA dissolution and re-precipitation) for triple oxygen isotope measurement. Detailed processing protocols can be found in refs. 43,45,71. Pre-treatment experiments were conducted at both Louisiana State University and China University of Geosciences (Wuhan).

### Isotope measurements
Sulfate $\delta^{34}S$ was measured through in-line combustion of ~0.3 mg of powdered $BaSO_4$ mixed with ~1.0 mg of $V_2O_5$ in a Flash elemental analyzer coupled to a Thermo Fisher Scientific Delta V Plus isotope-ratio mass spectrometer. Sulfate $\delta^{18}O$ was measured via CO gas converted from ~0.20 mg of $BaSO_4$ powder using a Thermal Conversion Elemental Analyzer at 1410 °C coupled to a Thermo Fisher Scientific Delta V Plus isotope-ratio mass spectrometer in continuous-flow mode. Sulfate $\Delta'^{17}O$ was determined using a Thermo MAT253 Plus

isotope-ratio mass spectrometer in dual-inlet mode via measuring $O_2$ generated offline from ~8 to 12 mg of pure $BaSO_4$ powder reacting with $BrF_5$, utilizing a $CO_2$ laser-fluorination system (further details refer to descriptions provided in refs. 32,47). The sulfur and oxygen isotope compositions are reported in δ-notation as per mil (‰) relative to Vienna Cañon Diablo Troilite (V-CDT) and Vienna Standard Mean Ocean Water (V-SMOW), respectively. The $δ^{34}S$ was calibrated using standards of two international (NBS127, 20.3‰, IAEA-SO-5, 0.5‰) and one inter-laboratory (OASIC-S, 14.5‰), while the $δ^{18}O$ using standards of one international (NBS127, 8.6‰) and one inter-laboratory (OASIC-O, 12.3‰). The analytical uncertainties (1σ) were better than ±0.1‰ and ±0.3‰ for $δ^{34}S$ and $δ^{18}O$, respectively. The standard deviation (1σ) for the $Δ^{'17}O$ is ±0.02‰ based on multiple runs ($N = 4$) of the same $BaSO_4$ sample. All isotope analyses were conducted in the International Center for Isotope Effects Research (ICIER) at Nanjing University.

## Data availability
All data generated or analysed during this study are included in this published article and its supplementary information files.

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

## Acknowledgements

We thank Bing Shen, Ruimin Wang, Matthew S. Dodd, Zihu Zhang, and Xinyang Chen for their help in field sampling and scientific discussion. Financial support is provided by the National Key Research and Devel-opment Program of China 2022YFF0800100 (C.L.), 2022YFF0800303 (Y.P.), National Natural Science Foundation of China Grant 42103072 (H.W.), 42173001 (Y.P.), 41825019, 42130208 (C. L.), 42173002 (X.C.), 42072335 (M.C.), the Fundamental Research Funds for the Central Universities 0206/14380150, 0206/14380185, 0206/14380174 (Y.P. and X.C.), China Postdoctoral Science Foundation 2021M691495 (H.W.), and Nanjing University Start-up fund (H.B.).

## Author contributions

C.L., Y.P., H.W., and H.B. designed the research. H.W. and Y.P. performed experiments, isotopic analyses and model simulations. All authors contributed to data interpretation. H.W. and H.B. wrote the manuscript with contributions from Y.P., C.L., and input from X.C. and M.C.

## Competing interests

The authors declare no competing interests.
