## [Peer Review File · Nature Communications]

Sulfate triple-oxygen-isotope evidence confirms oceanic oxygenation 570 million years agoReviewer #1 (Remarks to the Author):

Review of Wang et al., NatComm.

Wang et al., have made a remarkable discovery in strata bearing the Shuram excursion. The authors find triple oxygen isotope values ($\Delta^{17}O$) values down to ≈ -0.5 per mil in carbonate associated sulfate (CAS). Incredibly, these findings appear to be global in nature with the authors finding similar values across three separate paleocontinents. Moreover, the authors also find co-eval trends in $\delta^{34}S$, $\delta^{18}O$ and $\delta^{13}C$ values across their studied sections.

This discovery is important as, with the exception of post-Marinoan Snowball Earth deposits, it has been questioned whether CAS would ever bear large anomalous $\Delta^{17}O$ signatures, as it has been inferred that the seawater sulfate reservoir would reset such values toward the isotopic composition of water (i.e., 0). Therefore, I believe this work and this discovery will be of high impact to the community and of broad interest to the community at large and will motivate many future works to dig into the CAS $\Delta^{17}O$ record.

That said, I do find there are some issues with the current manuscript that I feel should be addressed before this work is fit for publication.

First this manuscript would benefit from a thorough edit. There are many grammatical errors, and odd wordings which I think detract from the importance of this work. I suggest a thorough read through before publication.

Second, the authors argue for the ventilation of H₂S rich waters to embed an atmospheric $\Delta^{17}O$ signature into sulfate. This suggestion has also been put forward to explain post-Snowball Earth. While it is almost a certainty that the $\Delta^{17}O$ composition of atmospheric O₂ across the Shuram was different to the present-day value of -0.5 (from the authors notes in the paper and the values of the model presumably it is somewhere between -10 and -2.3; Balci et al., 2007; Kohl & Bao, 2011), today this mechanism should produce some sulfate with a value of ≈ -0.15 to -0.025. However, it is unclear if this process actually occurs today, and whether any environment on the modern Earth is preserving such signatures. The authors need to provide further evidence that direct H₂S oxidation by O₂ has been shown to occur somewhere.

Third, this discovery does have implications for the interpretation of $\Delta^{17}O$ signatures deposited in the aftermath of the Marinoan Snowball Earth. The authors state in the manuscript that 'After all, the SE is only ~ 60 Myr younger than the basal Ediacaran $\Delta^{17}O$ -depletion event and the Earth system may still share similar responses and sensitivities to redox perturbations'. Despite the fact that I find this statement a bit non-sensical, if one takes this statement at face value it begs the question whether the post-Marinoan conditions were more like the Shuram or vice versa. In the former, the results of the present study question one of the most critical lines of evidence in support of a hard-Snowball state as the atmospheric conditions suggested to produce these values are unique to a Snowball Earth climate state with limited air sea exchange. Alternatively, was the Shuram more like the Snowball and for some reason produced extreme atmospheric signatures. Or are these more like the older mid-Proterozoic Sibley sulfates, low productivity, high CO₂, low O₂ and preserved in a much more restricted setting. The authors should discuss these possibilities.

Fourth, the authors (in my view) provide quite compelling evidence that much of the geochemistry of carbonates within the Shuram Excursion must be primary. To preserve the $\Delta^{17}O$ signals in CAS it suggests limited diagenesis of these units. The authors should be clear in the concluding paragraphs about what models of the Shuram Excursion this study supports and which ones it does not.

Fifth, the authors appear to dismiss the possibility of re-deposited evaporites bearing a large negative $\Delta^{17}O$ signatures. However, their arguments in the manuscript come across as a bit non-sensical. As evaporites have low preservation potential, just because there are few examples of large negative signatures preserved today does not mean that we haven't lost many of these deposits to weathering over the past 550 Myrs. That is, it is very hard to discount old evaporite weather especially since some old deposits have values consistent with what the authors find in

their CAS record. Also, it is worth noting that there are a number of large negative $\Delta 17O$ occurrences in the pre-Cryogenian record (Crockford et al., 2019a, Hodgskiss et al., 2019). Finally, is it possible that the extreme weathering brought on by Cryogenian glaciations may have actually unearthed some of these older deposits? Moreover, perhaps the authors should also consider other possibilities that have been put forward for Marinoan barites (e.g., subterranean groundwater discharge), ocean de-stratification etc (Crockford 2019b).

I hope these suggestions help. Again, this is an extremely important discovery that the authors have found and I am very supportive of this work for publication.

Reviewer #2 (Remarks to the Author):

Wang and co-authors present a study where carbonate associated sulfate (CAS) from 3 stratigraphic sections that represent the Shuram carbon isotope excursion (SE) was measured for triple oxygen isotope compositions. The study found that both the $\delta^{13}C$, $\delta^{18}O$, and $\Delta 17O$ values all shift more negative during the nadir of the SE. They conclude that the $\Delta 17O$ value represents air O_2 and that deep water oxidation with dissolved O_2 (that has the same triple oxygen isotope composition as atmospheric O_2) caused the SE. The manuscript is extremely well-written and tells a compelling story. There are a few major comments that should be addressed before publication.

- 1) The study relies heavily on the covariation of $\delta^{13}C$ of carbonate and the $\delta^{18}O$ and $\Delta 17O$ of the CAS. Would the $\delta^{13}C$ of the carbonate and the $\delta^{18}O$ of the sulfate be expected to covary in the way shown in Fig. 1? The $\delta^{18}O$ of modern atmospheric O_2 is $\sim +24$ ‰ and the $\Delta 17O$ is ~ -0.43 ‰ (Pack, 2021). If sulfate is taking on DO , wouldn't the $\delta^{18}O$ of the sulfate increase while the $\delta^{13}C$ of the carbonate decrease (due to the oxidation of organic matter with low $\delta^{13}C$ values)? Yet in Fig. 1 the $\delta^{18}O$ of the CAS is decreasing.
- 2) The $\Delta 17O$ of the sulfate is ~ -0.5 ‰ at the nadir. What might have caused the $\Delta 17O$ value of the CAS to be lower than the DO ?
- 3) The carbon isotope values are from the carbonate. If the $\delta^{13}C$ values are primary, they should not co-vary with the $\delta^{18}O$ values of the carbonate. The $\delta^{13}C$ value would be negative due to the light organic matter and the $\delta^{18}O$ value of the carbonate would remain relatively unchanged due to the oxygen isotope value coming from the seawater. However, in the Wonoka and Doushantuo Formations, the $\delta^{13}C$ and $\delta^{18}O$ values tend to covary (Derry, 2010). Since the manuscript weighs most of its conclusions on the covariation of the $\delta^{13}C$ value of carbonates with the triple oxygen isotope composition of the sulfate, the covariation with the $\delta^{18}O$ value of the carbonate should not just be ignored. I agree that a pure diagenesis signal that is of the same magnitude globally sounds unlikely based on modern ocean systems, yet a huge pool of unoxidized organic matter that experienced sustained slow oxidation over 7-8 Ma also seems unlikely based on modern ocean systems. At minimum, the manuscript should incorporate the $\delta^{18}O$ values of the carbonate that the sulfate was extracted from so that the entire geochemical picture can be critically examined.

Therefore, although the manuscript is well written and compelling, until these major general comments are addressed, the scientific validity of the conclusions cannot be evaluated. However, the manuscript is of wide interest and fitting for publication in Nature Communications. The review recommends the manuscript is revised and resubmitted before acceptance.

Minor comments:

Line 84: Aquatic photorespirations appears to have an intermediate value of about -0.2 ‰ (Barkan and Luz, 2011).

Line 174: What is 'spatial heterogeneity' refer to? The preceding section mentioned how homogenous the values are across continents, yet now it says there is spatial heterogeneity. Not clear what is being discussed.

References

- Barkan E. and Luz B. (2011) The relationships among the three stable isotopes of oxygen in air, seawater and marine photosynthesis. *Rapid Communications in Mass Spectrometry* 25, 2367-2369.
- Derry L. A. (2010) A burial diagenesis origin for the Ediacaran Shuram–Wonoka carbon isotope

anomaly. *Earth and Planetary Science Letters* 294, 152-162.
Pack A. (2021) Isotopic Traces of Atmospheric O₂ in Rocks, Minerals, and Melts. *Reviews in Mineralogy and Geochemistry* 86, 217-240.

Reviewer #3 (Remarks to the Author):

This manuscript presents sulfur and oxygen isotope profiles of the Shuram excursion at three different sections. The new data show a consistent pattern of ¹⁷O anomaly. Based on the coupled C, O, and S isotope patterns, the authors propose that the Shuram excursion resulted from an oxygenation event with enhanced pyrite oxidation and ocean ventilation.

The manuscript contains lots of novel and insightful data. I suggest a major revision based on my concerns below.

MAJOR COMMENTS

1. The authors analyzed the Shuram excursion at three sections. I noticed that the third section (i.e., Shuiquan Fm at the Mochia-Khntuk section in Tarim basin, Fig. 1C in the manuscript) is actually based on an online preprint (Wang et al., 2021) that has not been peer-reviewed or formally published.

I am still not yet convinced that the Mochia-Khntuk section (Wang et al., 2021) is correlative with the Shuram excursion. In my view, this manuscript should not be published unless it can independently validate the correlation issue, which is a critical assumption of this study.

This manuscript lacks the basic introduction of the Mochia-Khntuk section. It is inappropriate for the authors to incorrectly cite the Wang et al. (2022)'s paper for the Mochia-Khntuk section. The Wang et al. (2022) paper does not contain any original data of the Mochia-Khntuk section.

I also noticed that the carbon isotope data of two sections (i.e., the Wonoka and Shuiquan formations) are missing in the online supplement. This is frustrating. The readers cannot compare these sections on their own. These carbon isotope data serve as the essential foundation of this study. Without these carbon isotope data being available, I do not think this manuscript should be published.

2. The language should be further improved. The current version is still not satisfactory. There are grammatical errors from time to time. Perhaps the senior author of this manuscript (e.g., Huiming Bao) could make more efforts in revising the language.

3. The organization should be further improved. The model part is quite lengthy and not easy to follow. I suggest the authors shorten the model section.

The focus of this manuscript should be the ¹⁷O anomaly signals. I suggest the authors provide more in-depth discussions about the background, origin, pathway, and preservation of ¹⁷O anomaly signals in a specific section.

4. I suggest the authors consider citing or commenting on the Wu et al. (2015) paper, which I think is relevant to this study.

MINOR COMMENTS

Line 16, Line 32: " \sim 574-567 Ma,"

These two Re-Os ages should always be followed by their corresponding uncertainty (i.e., 574.0 ± 4.7 Ma, 567.3 ± 3.0 Ma.). The Rooney et al. (2020) paper never shows these ages without providing their uncertainty.

Line 74: "A comparable analogue is the finding that the deposition of ¹³C-depleted calcite cement

and a rise of the sulfate concentration coincided with an episode of large sulfate $\delta^{17}\text{O}$ depletion at the aftermath of Marinoan glaciation”.

This sentence should be moved to the Discussion section, after the authors show their new data. Here the authors only need to present the hypothesis. It is redundant and distractive to discuss any other intervals here.

Line 88: This paragraph is too long. I suggest the authors split it into two shorter ones.

Line 113: “The NaCl-leached sulfate potentially represents a mixture of primary and secondary sulfate...”

Theoretically, primary sulfate cannot be released before the host carbonates are acidified. Why here secondary sulfate could be mixed with primary sulfate in NaCl-leached sulfate? Why NaCl could release primary sulfate here?

Line 117: “Often, sulfate from oxidation of sulfide minerals has lower $\delta^{34}\text{S}$ and $\delta^{18}\text{O}$ values than the primary seawater sulfate.”

If this sentence is not talking about the authors’ own results, please add citations here.

Line 157: “will”

The word “will” in the discussion section should all be replaced with “would”. I suggest the authors double-check this issue throughout the entire manuscript.

Line 175: “circum-SE”

This is not a commonly used word.

Line 191: “Atmospheric O_2 is the only known source compound that bears a negative $\Delta^{17}\text{O}$ value.”

Please add citations here.

Line 213: “which provides a new dimensional information”

The word “information” is an uncountable noun.

Line 247: “Let us then explore another possibility, i.e., an increased”...

The writing looks informal. The modeling section is a bit lengthy, with lots of numbers that are not previously introduced. It is not easy to follow.

Line 276: “linked to global cooling”

Global cooling? This “global cooling” comes out of nowhere. Please either delete it or discuss it in detail.

Line 280 “On the other hand, it begs the question if the atmosphere O_2 had been distinctly depleted in $\delta^{17}\text{O}$ throughout the Ediacaran Period and the high sulfate concentrations at the aftermath of Marinoan Snowball Earth and during the SE merely facilitated the preservation of the sulfate $\delta^{17}\text{O}$ record, or atmospheric O_2 was distinctly depleted in $\delta^{17}\text{O}$ only during the two episodes”.

This manuscript shows lots of novel $\delta^{17}\text{O}$ data. But I feel that the $\delta^{17}\text{O}$ data have not been deeply discussed. Could the authors offer an in-depth discussion on the origin, pathway, and preservation of the $\delta^{17}\text{O}$ anomaly.

Line 290: “28 samples from one of the Canyon-Shoulder sections (31°8'46"S, 138°32'3"E) in South Australia”

There are many sections in Jon Husson’s paper. Please tell the readers which exact section in Husson’s paper the authors analyzed in this study.

Line 290: “39 samples from the Mochia-Khutuk section (41°26'29"N, 87°51'47"E) in Tarim”

I am still not yet convinced that the Mochia-Khutuk section is the Shuram excursion. This manuscript should not be published unless it can independently demonstrate it. This critical piece of information cannot rely on an online preprint (Wang et al., 2021).

Line 293: citation 59

It is inappropriate for the authors to incorrectly cite the Wang et al. (2022)'s paper for the Mochia-Khntuk section. The Wang et al. (2022) paper does not contain any original data of the Mochia-Khntuk section.

Lines 294 to 300: Method

Please delete the "(a) (b) (c) (d)" in this paragraph. This paragraph should be rewritten. This method section looks like a lab manual.

Lines 305-311: Method

Please delete the "(1) (2) (3)" in this paragraph. This method section looks like a lab manual.

Figure 1: I suggest the authors plot data of different sections along the same X-axis. I noticed that the authors stretched or squeezed the X-axis when plotting the O and S isotope data. It is difficult for me to compare.

Figure 1: The Pre-SE interval in South Australia does not show any O and S isotope data. Could the authors fill this gap?

Figure 1: Please explain the meaning of the dash lines in this figure. Is it running average?

Figure S3: Please explain the meaning of the dash lines in this figure. Some dash lines look arbitrary.

Line 490: "filled with a graduated color denote"

Gradual, not graduated.

REFERENCES

Rooney, A.D., Cantine, M.D., Bergmann, K.D., Gómez-Pérez, I., Al Baloushi, B., Boag, T.H., Busch, J.F., Sperling, E.A., Strauss, J.V., 2020. Calibrating the coevolution of Ediacaran life and environment. *Proceedings of the National Academy of Sciences*, 117, 16824–16830. <https://doi.org/10.1073/pnas.2002918117>.

Wang, R., Shen, B., Lang, X., Wen, B., Ma, H., Yin, Z., Peng, Y., Liu, Y., Zhou, C., 2021. A 20 million-year Great Ediacaran Glaciation witnessed the rise of the earliest animals. *Research Square* <https://doi.org/10.21203/rs.3.rs-793746/v1>.

Wang, Y., Chen, D., Liu, M., Liu, K., Tang, P., 2022. Ediacaran carbon cycling and Shuram excursion recorded in the Tarim Block, northwestern China. *Precambrian Research*, 377, 106694. <https://doi.org/10.1016/j.precamres.2022.106694>.

Wu, N., Farquhar, J., Fike, D.A., 2015. Ediacaran sulfur cycle: Insights from sulfur isotope measurements ($\Delta^{33}\text{S}$ and $\delta^{34}\text{S}$) on paired sulfate-pyrite in the Huqf Supergroup of Oman. *Geochimica et Cosmochimica Acta*, 164, 352–364. <https://doi.org/10.1016/j.gca.2015.05.031>.

Response to Reviewers' Comments

Reviewer #1 (Remarks to the Author):

Wang et al., have made a remarkable discovery in strata bearing the Shuram excursion. The authors find triple oxygen isotope values ($\Delta^{17}\text{O}$) values down to ≈ -0.5 per mil in carbonate associated sulfate (CAS). Incredibly, these findings appear to be global in nature with the authors finding similar values across three separate paleocontinents. Moreover, the authors also find co-eval trends in $\delta^{34}\text{S}$, $\delta^{18}\text{O}$ and $\delta^{13}\text{C}$ values across their studied sections.

This discovery is important as, with the exception of post-Marinoan Snowball Earth deposits, it has been questioned whether CAS would ever bear large anomalous $\Delta^{17}\text{O}$ signatures, as it has been inferred that the seawater sulfate reservoir would reset such values toward the isotopic composition of water (i.e., 0). Therefore, I believe this work and this discovery will be of high impact to the community and of broad interest to the community at large and will motivate many future works to dig into the CAS $\Delta^{17}\text{O}$ record.

That said, I do find there are some issues with the current manuscript that I feel should be addressed before this work is fit for publication.

Response: We thank the reviewer for the constructive comments, all of which have been addressed in the revised manuscript (see below).

First this manuscript would benefit from a thorough edit. There are many grammatical errors, and odd wordings which I think detract from the importance of this work. I suggest a thorough read through before publication.

Response: Thanks for bringing this to our attention. The manuscript has been carefully edited.

Second, the authors argue for the ventilation of H_2S rich waters to embed an atmospheric $\Delta^{17}\text{O}$ signature into sulfate. This suggestion has also been put forward to explain post-Snowball Earth. While it is almost a certainty that the $\Delta^{17}\text{O}$ composition

of atmospheric O₂ across the Shuram was different to the present-day value of -0.5‰ (from the authors notes in the paper and the values of the model presumably it is somewhere between -10 and -2.3; Balci et al., 2007; Kohl & Bao, 2011), today this mechanism should produce some sulfate with a value of ≈-0.15 to -0.025‰. However, it is unclear if this process actually occurs today, and whether any environment on the modern Earth is preserving such signatures. The authors need to provide further evidence that direct H₂S oxidation by O₂ has been shown to occur somewhere.

Response: Laboratory experiments have demonstrated that H₂S can be directly oxidized by O₂ (Chen and Morris, 1972; Oba and Poulson, 2009; Siang et al., 2017), with the chemical reaction being exothermic. While the exact pathways of this reaction are not fully understood and there are some unresolved steps, it is widely believed to occur in various natural environments where both compounds (H₂S and O₂) are present, such as sulfur springs and geysers (Xu et al., 1998), anoxic lakes (Gingras et al., 2011), swamps and wetlands (Wu et al., 2011), or marine sulfide-rich environments (e.g., the Black Sea; Neretin et al., 2001). We have included additional discussion about modern H₂S oxidation by O₂ in the revised manuscript, see lines 236-238.

Third, this discovery does have implications for the interpretation of $\Delta^{17}\text{O}$ signatures deposited in the aftermath of the Marinoan Snowball Earth. The authors state in the manuscript that ‘After all, the SE is only ~60 Myr younger than the basal Ediacaran ^{17}O -depletion event and the Earth system may still share similar responses and sensitivities to redox perturbations. Despite the fact that I find this statement a bit nonsensical, if one takes this statement at face value it begs the question whether the post-Marinoan conditions were more like the Shuram or vice versa. In the former, the results of the present study question one of the most critical lines of evidence in support of a hard-Snowball state as the atmospheric conditions suggested to produce these values are unique to a Snowball Earth climate state with limited air sea exchange. Alternatively, was the Shuram more like the Snowball and for some reason produced extreme atmospheric signatures. Or are these more like the older mid-Proterozoic Sibley sulfates, low productivity, high CO₂, low O₂ and preserved in a much more restricted setting.

The authors should discuss these possibilities.

Response: Thanks for these comments. From your comments, we can tell that the statement of *'After all, the SE is only ~60 Myr ...'* mentioned above has achieved its goal: getting reader to ponder deeply the origin of the ^{17}O -depletion in air O_2 in geological history. Meanwhile, we also felt the statement can generate misunderstanding. We have, therefore, discarded the statement in the revised manuscript. Regarding the reviewer's question about whether the ^{17}O data of the post-Marinoan and Shuram can provide insight into the similarity or differences between the two events, in the original concluding paragraph we had posted two possible scenarios: *"... if the atmosphere O_2 had been distinctly depleted in ^{17}O throughout the Ediacaran Period and the high sulfate concentrations at the aftermath of Marinoan Snowball Earth and during the SE merely facilitated the preservation of the sulfate ^{17}O record, or if atmospheric O_2 was distinctly depleted in ^{17}O only during the two episodes"* (see lines 309-315 in the revised manuscript). To resolve these possibilities, more Ediacaran ^{17}O data coverage is needed. Similarly, further research is required to answer if the Mesoproterozoic Sibley evaporite sulfate's ^{17}O -depletion shares a common or different biosphere-atmospheric background with the Shuram or the basal-Ediacaran ones. We are now very excited by the many new research opportunities opened up by this Shuram discovery, as the reviewer aptly appreciated.

Fourth, the authors (in my view) provide quite compelling evidence that much of the geochemistry of carbonates within the Shuram Excursion must be primary. To preserve the $\Delta^{17}\text{O}$ signals in CAS it suggests limited diagenesis of these units. The authors should be clear in the concluding paragraphs about what models of the Shuram Excursion this study supports and which ones it does not.

Response: Good suggestion. We are convinced our data provide quite compelling evidence for the primary nature of the Shuram Excursion. A clearer statement has been added in the concluding paragraph. See lines 324-327.

Fifth, the authors appear to dismiss the possibility of re-deposited evaporites bearing a

large negative $\Delta^{17}\text{O}$ signatures. However, their arguments in the manuscript come across as a bit non-sensical. As evaporites have low preservation potential, just because there are few examples of large negative signatures preserved today does not mean that we haven't lost many of these deposits to weathering over the past 550 Myrs. That is, it is very hard to discount old evaporite weather especially since some old deposits have values consistent with what the authors find in their CAS record. Also, it is worth noting that there are a number of large negative $\Delta^{17}\text{O}$ occurrences in the pre-Cryogenian record (Crockford et al., 2019a, Hodgskiss et al., 2019). Finally, is it possible that the extreme weathering brought on by Cryogenian glaciations may have actually unearthed some of these older deposits? Moreover, perhaps the authors should also consider other possibilities that have been put forward for Marinoan barites (e.g., subterranean groundwater discharge), ocean destratification etc (Crockford 2019b).

Response: We understand the reviewer's concern regarding the potential role of old evaporites as the sulfate source for the SE oceans. However, it is unlikely that old evaporites serve as a significant source of the SE sulfate due to several reasons.

(1) Our CAS data from three examined paleocontinents all show distinctly negative $\Delta^{17}\text{O}$ values and similar $\delta^{34}\text{S}$ and $\delta^{18}\text{O}$ values during the SE. This observation argues against evaporite dissolution being the main source of the added sulfate input, as contribution from evaporite dissolution should be local and it is unlikely that evaporites with the same multi-isotope composition would exist on different continents.

(2) The positive correlations between $\Delta^{17}\text{O}$ and $\delta^{18}\text{O}$, especially in South China and South Australia, suggest a mixture between two end-member sulfate pools, one with a higher $\delta^{18}\text{O}$ and a close-to-zero $\Delta^{17}\text{O}$ value and the other with lower $\delta^{18}\text{O}$ and negative $\Delta^{17}\text{O}$ values. The end-member sulfate source could be then estimated by the cross-plots of $\Delta^{17}\text{O}$ and $\delta^{18}\text{O}$, which, however, do not support the pre-existing (i.e., 1.4-Ga and 1.7-Ga; Crockford et al., 2019) gypsum with significant negative ^{17}O anomalies as one of the end-member sulfate pools. Note that all data for the pre-Ediacaran gypsum that have large negative ^{17}O anomalies fall outside of the region of the potential end-member sulfate (see Figure 1 below, which we have also added to Supplementary Information

as Supplementary Fig. 4). (3) The pre-Ediacaran evaporites are all small in scale. The current pre-Ediacaran evaporite volume is $\sim 4.7 \times 10^5 \text{ km}^3$ in total (Evans, 2006). If being all gypsum and all dissolved, they could only supply $\sim 0.6 \text{ Myr}$ of the sulfate flux (i.e., $\sim 1 \times 10^{13} \text{ mol/yr}$) needed in Shields et al. (2019)'s model estimate. Although the current pre-Ediacaran evaporite volume is the residue after experiencing dissolution, these numbers require an unreasonably large volume of evaporites exposed in the pre-Ediacaran time, and evidence for massive pre-Ediacaran evaporite dissolution is lacking. We have added additional relevant discussion regarding these possibilities in the revised manuscript (see lines 220-232 and also Supplementary Fig. 4).

The reviewer entertained the idea that the extreme weathering brought on by Cryogenian glaciations may have unearthed some of the older deposits. This is likely, but we do not think it played a role for the SE event, given the quite long-time lag of $>60 \text{ Ma}$ between SE and the end of Cryogenian glaciations.

We agree with the reviewer that other possibilities, such as subterranean groundwater discharge and ocean destratification, that have been put forward for Marinoan barites may have played a role during the SE as well. We note that both these events are transient or short-termed by nature and are therefore cannot be entirely responsible for the ca. 6-7 Myrs sulfate $\Delta^{17}\text{O}$ negative excursion. In addition, groundwater discharge was invoked to supply large amounts of barium for the formation of the Marinoan Barite (Peter et al., 2019), and our CAS data should be independent of Ba cycling. On the other hand, when considering later diagenetic effect, we discussed a scenario in which later O_2 -rich water (e.g., groundwater) circulated through pyrite-rich carbonates and oxidized the pyrite to sulfate. This is a distinct possibility if the "late" water carried ^{17}O -depleted air O_2 . However, we have presented arguments that this scenario was less likely (see discussion at lines 176-189).

I hope these suggestions help. Again, this is an extremely important discovery that the authors have found and I am very supportive of this work for publication.

Response: We show our appreciation again for the reviewer's positive evaluation and valuable inputs.

Figure 1. Crossplots of the $\Delta^{17}\text{O}$ and $\delta^{18}\text{O}$ for Ediacaran CAS and for the pre-Ediacaran gypsum with significant negative ^{17}O anomalies. Strong positive correlations between $\Delta^{17}\text{O}_{\text{CAS}}$ and $\delta^{18}\text{O}_{\text{CAS}}$ in South China and South Australia suggest a mixture between two end-member sulfate pools, one with a higher $\delta^{18}\text{O}$ and a close-to-zero $\Delta^{17}\text{O}$ value, and the other with a lower $\delta^{18}\text{O}$ and a negative $\Delta^{17}\text{O}$ value. The sulfate source associated with the latter pool, represented by the five-pointed stars, can be estimated based on the correlation trends of $\Delta^{17}\text{O}$ and $\delta^{18}\text{O}$. Notably, the data for pre-existing (i.e., 1.4-Ga and 1.7-Ga) gypsum with significant negative ^{17}O anomalies do not align with the region defined by the red and blue dashed lines as well as the surrounding grey shaded area, which represent isotope composition of the potential end-member sulfate. Therefore, it does not support pre-Ediacaran gypsum as one of the end-member sulfate pools.

Reviewer #2 (Remarks to the Author):

Wang and co-authors present a study where carbonate associated sulfate (CAS) from 3 stratigraphic sections that represent the Shuram carbon isotope excursion (SE) was measured for triple oxygen isotope compositions. The study found that both the $\delta^{13}\text{C}$, $\delta^{18}\text{O}$, and $\Delta^{17}\text{O}$ values all shift more negative during the nadir of the SE. They conclude that the $\Delta^{17}\text{O}$ value represents air O_2 and that deep water oxidation with dissolved O_2 (that has the same triple oxygen isotope composition as atmospheric O_2) caused the SE. The manuscript is extremely well-written and tells a compelling story. There are a few major comments that should be addressed before publication.

1) The study relies heavily on the covariation of $\delta^{13}\text{C}$ of carbonate and the $\delta^{18}\text{O}$ and $\Delta^{17}\text{O}$ of the CAS. Would the $\delta^{13}\text{C}$ of the carbonate and the $\delta^{18}\text{O}$ of the sulfate be expected to covary in the way shown in Fig. 1? The $\delta^{18}\text{O}$ of modern atmospheric O_2 is $\sim +24\text{‰}$ and the $\Delta^{17}\text{O}$ is $\sim -0.43\text{‰}$ (Pack, 2021). If sulfate is taking on DO, wouldn't the $\delta^{18}\text{O}$ of the sulfate increase while the $\delta^{13}\text{C}$ of the carbonate decrease (due to the oxidation of organic matter with low $\delta^{13}\text{C}$ values)? Yet in Fig. 1 the $\delta^{18}\text{O}$ of the CAS is decreasing.

Response: Thanks for the comment. However, a few uncertainties prevent us from anticipating that sulfate $\delta^{18}\text{O}$ would increase while the carbonate $\delta^{13}\text{C}$ decrease. Yes, increased aerobic organic matter mineralization (and oxidation of the reduced sulfur species) would decrease the carbonate $\delta^{13}\text{C}$ value but may not increase the sulfate's $\delta^{18}\text{O}$ value. That is because the $\delta^{18}\text{O}$ difference between sulfate produced during aerobic and anaerobic oxidation could be less than 2‰ (9.5‰ vs. $\sim 8.0\text{‰}$ in Balci et al., 2012) today. This is due to the fact that only a small fraction ($<25\%$) of the oxygen in product sulfate is derived from O_2 even under aerobic conditions. Another factor is the kinetic isotope effect ($\text{KIE}_{\text{O}_2 \rightarrow \text{SO}_4}$) of ~ 0.985 (Cao and Bao, 2021), which attenuated the relatively high $\delta^{18}\text{O}$ value of O_2 when being incorporated into product sulfate. Finally, we are not sure what the $\delta^{18}\text{O}$ of air O_2 was during the Shuram; the value may well be much lighter than today's 24‰.

Our interpretation is that the decrease in sulfate $\delta^{18}\text{O}$ along stratigraphy reflects an

isotopic signature of a mixture with a changing mixing ratio. The S- and O-isotope composition tells us that sulfate is composed of two end-members: a newly-formed sulfate from sulfide oxidation and a residual sulfate that has undergone MSR (See Supplementary Fig. 4 we added in Supplementary Information). The $\delta^{18}\text{O}$ value of the former is relatively low, whereas the residual sulfate has a relatively high oxygen isotope composition due to Rayleigh process and due to exchange with water oxygen toward an equilibrium value. We interpret the decrease in sulfate $\delta^{18}\text{O}$ along stratigraphy as a result of increased sulfate concentrations during the SE, which are supported not only by our data (see discussion in lines 205-217), but also independently by previous studies (e.g., Fike et al., 2006; Kaufman et al., 2007; McFadden et al., 2008; Shi et al., 2018).

2) The $\Delta^{17}\text{O}$ of the sulfate is $\sim -0.5\text{‰}$ at the nadir. What might have caused the $\Delta^{17}\text{O}$ value of the CAS to be lower than the DO?

Response: We don't think the $\Delta^{17}\text{O}$ value of the CAS can be lower than that of the DO. We may have not expressed it clearly in the text, but this is the Ediacaran. The Precambrian atmosphere is widely believed to have been different from today's and air O_2 could have had significantly more negative $\Delta^{17}\text{O}$ values due to higher CO_2/O_2 ratios and/or different levels of bioproductivity (Cao and Bao, 2013). In fact, sulfate $\Delta^{17}\text{O}$ values as low as -1.64‰ have been measured from Neoproterozoic glacial sediments, from which extremely anomalous $\Delta^{17}\text{O}$ values between -42‰ and -6.6‰ were inferred for the contemporaneous atmospheric O_2 . Our model results suggest that, if we assume that 1/4 of the oxygen in the fresh sulfate ($\Delta^{17}\text{O} = -0.75\text{‰}$; see Supplementary Fig. 7) comes from atmospheric O_2 , the $\Delta^{17}\text{O}(\text{O}_2)$ value during the SE should be much more negative than -3‰ , not the today's $\sim -0.5\text{‰}$.

3) The carbon isotope values are from the carbonate. If the $\delta^{13}\text{C}$ values are primary, they should not co-vary with the $\delta^{18}\text{O}$ values of the carbonate. The $\delta^{13}\text{C}$ value would be negative due to the light organic matter and the $\delta^{18}\text{O}$ value of the carbonate would remain relatively unchanged due to the oxygen isotope value coming from the seawater.

However, in the Wonoka and Doushantuo Formations, the $\delta^{13}\text{C}$ and $\delta^{18}\text{O}$ values tend to covary (Derry, 2010). Since the manuscript weighs most of its conclusions on the covariance of the $\delta^{13}\text{C}$ value of carbonates with the triple oxygen isotope composition of the sulfate, the covariation with the $\delta^{18}\text{O}$ value of the carbonate should not just be ignored. I agree that a pure diagenesis signal that is of the same magnitude globally sounds unlikely based on modern ocean systems, yet a huge pool of unoxidized organic matter that experienced sustained slow oxidation over 7-8 Ma also seems unlikely based on modern ocean systems. At minimum, the manuscript should incorporate the $\delta^{18}\text{O}$ values of the carbonate that the sulfate was extracted from so that the entire geochemical picture can be critically examined.

Response: Thanks for these comments. The $\delta^{18}\text{O}$ values of the carbonate of the three sections and related discussion have been included (see Supplementary Data 1 and lines 190-204 in revised manuscript), but some statements require clarification.

1) While the $\delta^{13}\text{C}$ and $\delta^{18}\text{O}$ values appear to covary during SE in the Doushantuo Formation at Jiulongwan, this is not the case for much of the other sections in South China, such as the Sishang, Dongdahe, Jijiawan, and Miaohe, as reported by (Lu et al., 2013). Therefore, it is unlikely that the mechanism responsible for $\delta^{18}\text{O}_{\text{carb}}$ shift was the same as for the Shuram ^{13}C excursion, and vice versa. The interpretation of the apparent $\delta^{13}\text{C}$ and $\delta^{18}\text{O}_{\text{carb}}$ covariation may not be uniquely diagenetic, and the correlation may not be caused by a single process. The decrease of $\delta^{18}\text{O}_{\text{carb}}$ could be linked to higher temperature, drop in the $\delta^{18}\text{O}$ of ambient water, or diagenesis. A commonly used argument is that the $\delta^{18}\text{O}_{\text{carb}}$ is more susceptible to resetting than the corresponding $\delta^{13}\text{C}_{\text{carb}}$ due to the greater abundance of external oxygen compared to carbon during later fluid-rock interactions. Therefore, the $\delta^{13}\text{C}_{\text{carb}}$ could be of primary origin even if the $\delta^{18}\text{O}_{\text{carb}}$ is affected by diagenesis. Many studies have been conducted to specifically explore the co-variation of $^{13}\text{C}_{\text{carb}}$ and $\delta^{18}\text{O}_{\text{carb}}$ in the SE carbonates, yet this remains an open question (e.g., Derry, 2010; Grotzinger et al., 2011; Knauth and Kennedy, 2009; Tahata et al., 2013). Our study found that sulfur and triple oxygen isotope composition in sulfate co-vary with the $^{13}\text{C}_{\text{carb}}$ excursion, supporting the SE as a primary event. This

finding seemingly underscores the necessity to explore primary processes that can explain the co-variation of $\delta^{13}\text{C}_{\text{carb}}$ and $\delta^{18}\text{O}_{\text{carb}}$, but at this time our report cannot delve deeper into this topic as it requires more data and goes beyond the scope of the research. Please see the added discussion at lines 190-204 in the revised manuscript.

2) Our modeling, based on sulfur cycling, demonstrates that an enhanced oxidation of the pre-existing reduced sulfur pool lasting for only ~ 1.5 Myrs can explain the sulfate sulfur and oxygen isotopes. However, the sulfate isotopes are closely associated with the carbonate $\delta^{13}\text{C}$, which exhibits a negative excursion lasting ~ 7 Myrs (i.e., 574-567 Ma). This suggests that it may not be necessary to propose a sustained, slow oxidation of organic matter over 7-8 Ma to account for the SE event. Nevertheless, additional models, such as those focused on surface carbon cycling, would be required to reconstruct the driving forces and responses of seawater DIC $\delta^{13}\text{C}$, which falls beyond the scope of our study.

Therefore, although the manuscript is well written and compelling, until these major general comments are addressed, the scientific validity of the conclusions cannot be evaluated. However, the manuscript is of wide interest and fitting for publication in Nature Communications. The review recommends the manuscript is revised and resubmitted before acceptance.

Response: We appreciate the insightful and constructive feedback provided by the reviewer for our manuscript.

Minor comments:

Line 84: Aquatic photorespirations appears to have an intermediate value of about -0.2‰ (Barkan and Luz, 2011).

Response: We may misunderstand the reviewer's meaning. In general, O_2 generated by photosynthesis, terrestrial or aquatic, sources its oxygen entirely from the ambient water. As long as the seawater $\Delta^{17}\text{O}$ has not changed significantly over geological history, which should be true, then the sentence at Line 84 is valid (now it is at lines 83-85). We note that the photosynthetic O_2 has only slightly positive $\Delta^{17}\text{O}$ values ($+0.026\text{‰}$) with

respect to the substrate water (Barkan and Luz, 2011). Modern seawater $\Delta^{17}\text{O}$ is measured to be -0.005‰ (with respect to VSMOW; the slope is 0.528; Luz and Barkan, 2010). Therefore, the $\Delta^{17}\text{O}$ of the photosynthetic O_2 would be only about $+0.02\text{‰}$.

Line 174: What is 'spatial heterogeneity' refer to? The preceding section mentioned how homogenous the values are across continents, yet now it says there is spatial heterogeneity. Not clear what is being discussed.

Response: Thanks. We could have expressed it better. A minor revision has been made to enhance the clarity of this statement (see lines 205-207). Spatial heterogeneity refers to geographic or depth heterogeneity of sulfate's sulfur and triple oxygen isotope composition. Such heterogeneity is expected when seawater sulfate concentration is low, such as during the pre- or post-SE interval.

Reviewer #3 (Remarks to the Author):

This manuscript presents sulfur and oxygen isotope profiles of the Shuram excursion at three different sections. The new data show a consistent pattern of ^{17}O anomaly. Based on the coupled C, O, and S isotope patterns, the authors propose that the Shuram excursion resulted from an oxygenation event with enhanced pyrite oxidation and ocean ventilation.

The manuscript contains lots of novel and insightful data. I suggest a major revision based on my concerns below.

MAJOR COMMENTS

1. The authors analyzed the Shuram excursion at three sections. I noticed that the third section (i.e., Shuiquan Fm at the Mochia-Khntuk section in Tarim basin, Fig. 1C in the manuscript) is actually based on an online preprint (Wang et al., 2021) that has not been peer-reviewed or formally published.

Response: The reviewer is correct. The latest update is that Wang et al.'s paper has been accepted for publication by the Journal 'National Science Review' and is online now (it is cited as Wang et al., 2023 in the revised version).

Reference: Wang, R., Shen, B., Lang, X., Wen, B., Mitchell, R.N., Ma, H., Yin, Z., Peng, Y., Liu, Y., Zhou, C., 2023. A Great late Ediacaran ice age. *Natl. Sci. Rev.* 12, 4683–4698. <https://doi.org/10.1093/nsr/nwad117>.

I am still not yet convinced that the Mochia-Khutuk section (Wang et al., 2021) is correlative with the Shuram excursion. In my view, this manuscript should not be published unless it can independently validate the correlation issue, which is a critical assumption of this study.

Response: We appreciate the reviewer's concerns regarding the age constraints and correlation of the Shuiquan Formation at Mochia-Khutuk section. U-Pb ages from Ren et al. (2020) and Xu et al. (2009) have constrained the formation age to be between 615 ± 6 Ma and 541 ± 6 Ma, which supports the Ediacaran age of the succession. Additionally, Vendotaenid fossils have been reported from the Shuiquan Formation,

which are commonly found in post-Marinoan rocks (Xiao et al., 2004). We argue that the negative carbon isotope excursion with a nadir of -11.9‰ recorded in the Shuiquan Formation at Mochia-Khutuk section should be correlated to the Shuram excursion for the following reasons.

(1) No other long-duration, negative $\delta^{13}\text{C}_{\text{carb}}$ excursion (as low as -12‰) has been reported between 615 Ma and 541 Ma on any continent globally, except for the Shuram excursion or the equivalent Wonoka/DOUNCE/EN3. (2) The negative $\delta^{13}\text{C}$ excursion at Mochia-Khutuk section (NE Tarim) is comparable in magnitude, pattern of shift, lithology, and sedimentology to the largest $\delta^{13}\text{C}$ anomaly (lasting for >50 m with a nadir at ca. -12‰) observed in Aksu area, NW Tarim, which has been correlated with the SE based on multiple lines of evidence and precise dating constraints (Wang et al., 2022). (3) Other unique geochemical signals, e.g., the decoupling of $\delta^{13}\text{C}_{\text{carb}}$ and $\delta^{13}\text{C}_{\text{org}}$ and a concurrent decrease in $\delta^{18}\text{O}$ values, as commonly observed in other typical SE sections, were found in the Mochia-Khutuk section (Wang et al., 2023).

The Mochia-Khutuk section should be correlated to the Shuram excursion is also supported by a latest finding. A recent study (Dodd et al., 2023 accepted in *Nature*) measured carbonate-associated phosphate (CAP) of the Mochia-Khutuk section and found a “M-shape” pattern in CAP, a feature observed in global SE sections including the Jiulongwan (the same section as reported in our study) and Sishang sections of the Doushantuo Fm. from South China, the Cerro Rajón section of the Clemente, Pitiquito and Gamuza formations from northern Mexico, the Death Valley section of the Johnnie Fm. from southwestern USA, and the Parachilna Gorge section of the Wonoka Fm. (the same section as reported in our study) from South Australia.

We have extensively discussed this issue in the revised Supplementary Information and have cited the relevant references (Xiao et al., 2004; Wang et al., 2022 PR; Wang et al., 2023 NSR; Dodd et al., 2023). However, we have decided not to include much of the above discussions in the main text of our manuscript as they are not directly related to the topic of this study.

This manuscript lacks the basic introduction of the Mochia-Khutuk section. It is inappropriate for the authors to incorrectly cite the Wang et al. (2022)'s paper for the Mochia-Khntuk section. The Wang et al. (2022) paper does not contain any original data of the Mochia-Khntuk section.

Response: The reviewer is correct. The reference has been replaced by 'Wang et al., 2023 NSR', and also see our response above.

I also noticed that the carbon isotope data of two sections (i.e., the Wonoka and Shuiquan formations) are missing in the online supplement. This is frustrating. The readers cannot compare these sections on their own. These carbon isotope data service as the essential foundation of this study. Without these carbon isotope data being available, I do not think this manuscript should be published.

Response: Thanks for pointing out our omission. We now have included the carbon isotope data of the Wonoka and Shuiquan formations in Supplementary Data 1.

2. The language should be further improved. The current version is still not satisfactory. There are grammatical errors from time to time. Perhaps the senior author of this manuscript (e.g., Huiming Bao) could make more efforts in revising the language.

Response: Thanks for the comment. The full text was carefully revised.

3. The organization should be further improved. The model part is quite lengthy and not easy to follow. I suggest the authors shorten the model section.

Response: Thanks for the suggestion. We wanted to ensure that readers or other researchers can reproduce our model and results, hence the much details in the original manuscript. We have done some revisions and taken steps to shorten it by moving the following information to Supplementary Information (see model section at lines 246-315). *"The model considers both sulfate fluxes and changes in sulfate isotope composition over time, allowing $\delta^{34}\text{S}$ and $\delta^{18}\text{O}$ to be altered by microbial sulfur redox processes as prescribed in previous models. For the $\Delta^{17}\text{O}$, we assume the newly added sulfate from pyrite weathering on the continents and/or from aerobic oxidation of reduced sulfur species in the ocean carry ^{17}O anomalies, which are erased by sulfur*

redox processes through enhanced exchange of sulfate oxygen with the ambient water via a backward exchange between sulfite and sulfate and/or via anaerobic oxidation of H_2S/S^0 by nitrate or Fe(III) in water or sediments.”

The focus of this manuscript should be the ^{17}O anomaly signals. I suggest the authors provide more in-depth discussions about the background, origin, pathway, and preservation of ^{17}O anomaly signals in a specific section.

Response: We thank the reviewer’s comment. A specific section about origin, pathway, and preservation of ^{17}O anomaly signals in geological sulfate has been added in the revised manuscript, please see lines 110-130.

4. I suggest the authors consider citing or commenting on the Wu et al. (2015) paper, which I think is relevant to this study.

Response: This paper has been cited (No. 53), see line 185.

MINOR COMMENTS

Line 16, Line 32: “~574-567 Ma,”

These two Re-Os ages should always be followed by their corresponding uncertainty (i.e., 574.0 ± 4.7 Ma, 567.3 ± 3.0 Ma.). The Rooney et al. (2020) paper never shows these ages without providing their uncertainty.

Response: Revised, see line 35.

Line 74: “A comparable analogue is the finding that the deposition of ^{13}C -depleted calcite cement and a rise of the sulfate concentration coincided with an episode of large sulfate ^{17}O depletion at the aftermath of Marinoan glaciation”.

This sentence should be moved to the Discussion section, after the authors show their new data. Here the authors only need to present the hypothesis. It is redundant and distractive to discuss any other intervals here.

Response: Thanks. This sentence has been removed in the revised manuscript.

Line 88: This paragraph is too long. I suggest the authors split it into two shorter ones.

Response: Revised, see lines 89-95.

Line 113: “The NaCl-leached sulfate potentially represents a mixture of primary and secondary sulfate...”

Theoretically, primary sulfate cannot be released before the host carbonates are acidified. Why here secondary sulfate could be mixed with primary sulfate in NaCl-leached sulfate? Why NaCl could release primary sulfate here?

Response: To avoid misunderstanding, we have discarded this statement and revised this sentence to read as “Prior to the HCl-solution extraction, repeated NaCl-solution leaching can effectively remove most, if not all, of the syn-sedimentary evaporites (if present), sulfate generated by post-depositional processes (e.g., diagenesis, laboratory treatment), and modern atmospheric deposition.” (See lines 133-137).

Line 117: “Often, sulfate from oxidation of sulfide minerals has lower $\delta^{34}\text{S}$ and $\delta^{18}\text{O}$ values than the primary seawater sulfate.”

If this sentence is not talking about the authors’ own results, please add citations here.

Response: Added, see line 138.

Line 157: “will”

The word “will” in the discussion section should all be replaced with “would”. I suggest the authors double-check this issue throughout the entire manuscript.

Response: Thanks for pointing it, and all were revised.

Line 175: “circum-SE”

This is not a commonly used word.

Response: Modified as “pre- or post-SE” in the revised manuscript, see line 206.

Line 191: “Atmospheric O_2 is the only known source compound that bears a negative $\Delta^{17}\text{O}$ value.”

Please add citations here.

Response: Added, see lines 113-116.

Line 213: “which provides a new dimensional information”

The word “information” is an uncountable noun.

Response: The sentence has been revised, see lines 253-255.

Line 247: “Let us then explore another possibility, i.e., an increased”...

The writing looks informal. The modeling section is a bit lengthy, with lots of numbers that are not previously introduced. It is not easy to follow.

Response: Thanks for bringing this to our attention. The sentence has been revised (see lines 281-283). We have done some revisions on the modeling section (explanations for parameters were added, see lines 281-308) and shortened it by moving the following information to Supplementary Information. *“The model considers both sulfate fluxes and changes in sulfate isotope composition over time, allowing $\delta^{34}\text{S}$ and $\delta^{18}\text{O}$ to be altered by microbial sulfur redox processes as prescribed in previous models. For the $\Delta^{17}\text{O}$, we assume the newly added sulfate from pyrite weathering on the continents and/or from aerobic oxidation of reduced sulfur species in the ocean carry ^{17}O anomalies, which are erased by sulfur redox processes through enhanced exchange of sulfate oxygen with the ambient water via a backward exchange between sulfite and sulfate and/or via anaerobic oxidation of $\text{H}_2\text{S}/\text{S}^0$ by nitrate or Fe(III) in water or sediments.”* In addition, a comprehensive description of the model and its associated parameters can be found in the Supplementary Information.

Line 276: “linked to global cooling”

Global cooling? This “global cooling” comes out of nowhere. Please either delete it or discuss it in detail.

Response: Deleted.

Line 280 “On the other hand, it begs the question if the atmosphere O_2 had been distinctly depleted in ^{17}O throughout the Ediacaran Period and the high sulfate concentrations at the aftermath of Marinoan Snowball Earth and during the SE merely facilitated the preservation of the sulfate ^{17}O record, or atmospheric O_2 was distinctly depleted in ^{17}O only during the two episodes”.

This manuscript shows lots of novel ^{17}O data. But I feel that the ^{17}O data have not been deeply discussed. Could the authors offer an in-depth discussion on the origin, pathway, and preservation of the ^{17}O anomaly.

Response: Thanks for this comment. A specific section about origin, pathway, and preservation of ^{17}O anomaly signals in geological sulfate has been added in the revised

manuscript, please see lines 110-130.

Line 290: “28 samples from one of the Canyon-Shoulder sections (31°8'46"S, 138°32'3"E) in South Australia”

There are many sections in Jon Husson’s paper. Please tell the readers which exact section in Husson’s paper the authors analyzed in this study.

Response: Revised. It is the Parachilna Gorge section (31°9'51.6"S, 138°31'43.2"E) which is roughly equivalent to the section numbered ‘⑥’ reported in Husson et al. (2015).

Line 290: “39 samples from the Mochia-Khutuk section (41°26'29"N, 87°51'47"E) in Tarim”

I am still not yet convinced that the Mochia-Khutuk section is the Shuram excursion. This manuscript should not be published unless it can independently demonstrate it. This critical piece of information cannot rely on an online preprint (Wang et al., 2021).

Response: Revised, and see our Response to Comments of “I am still not yet convinced that the Mochia-Khutuk section (Wang et al., 2021) is correlative with...”, in which we present multiple lines of evidence for supporting the fact that the Shuram excursion is recorded in the Mochia-Khutuk section.

Line 293: citation 59

It is inappropriate for the authors to incorrectly cite the Wang et al. (2022)’s paper for the Mochia-Khntuk section. The Wang et al. (2022) paper does not contain any original data of the Mochia-Khntuk section.

Response: Changed.

94 to 300: Method

Please delete the “(a) (b) (c) (d)” in this paragraph. This paragraph should be rewritten. This method section looks like a lab manual.

Response: Deleted, and this paragraph has been revised at lines 339-349.

5-311: Method

Please delete the “(1) (2) (3)” in this paragraph. This method section looks like a lab

manual.

Response: Deleted.

Figure 1: I suggest the authors plot data of different sections along the same X-axis. I noticed that the authors stretched or squeezed the X-axis when plotting the O and S isotope data. It is difficult for me to compare.

Response: The figure has been modified following your suggestions.

1: The Pre-SE interval in South Australia does not show any O and S isotope data. Could the authors fill this gap?

Response: Good suggestion. We tried it, but the problem is that the “Wonoka” $\delta^{13}\text{C}_{\text{carb}}$ downturn is poorly preserved, or even non-existent in South Australia (see Husson et al., 2015).

Figure 1: Please explain the meaning of the dash lines in this figure. Is it running average?

Response: Thanks. The lines mean LOWESS (locally weighted scatterplot smoothing) curves, and this note has been added in the figure caption.

Figure S3: Please explain the meaning of the dash lines in this figure. Some dash lines look arbitrary.

Response: Added, the lines mean LOWESS curves in the revised figures.

Line 490: “filled with a graduated color denote”

Gradual, not graduated.

Response: Revised.

REFERENCES

- Balci, N., Mayer, B., Shanks, W.C., Mandernack, K.W., 2012. Oxygen and sulfur isotope systematics of sulfate produced during abiotic and bacterial oxidation of sphalerite and elemental sulfur. *Geochim. Cosmochim. Acta* 77, 335–351.
- Cao, X., Bao, H., 2021. Small Triple Oxygen Isotope Variations in Sulfate: Mechanisms and Applications. *Rev. Mineral. Geochemistry* 86, 463–488.
- Cao, X., Bao, H., 2013. Dynamic model constraints on oxygen-17 depletion in atmospheric O₂ after a snowball Earth. *Proc. Natl. Acad. Sci.* 110, 14546–14550.

- Chen, K.Y., Morris, J.C., 1972. Kinetics of oxidation of aqueous sulfide by oxygen. *Environ. Sci. Technol.* 6, 529–537.
- Crockford, P.W., Kunzmann, M., Bekker, A., Hayles, J., Bao, H., Halverson, G.P., Peng, Y., Bui, T.H., Cox, G.M., Gibson, T.M., Wörndle, S., Rainbird, R., Lepland, A., Swanson-Hysell, N.L., Master, S., Sreenivas, B., Kuznetsov, A., Krupenik, V., Wing, B.A., 2019. Claypool continued: Extending the isotopic record of sedimentary sulfate. *Chem. Geol.* 513, 200–225.
- Dodd., M. S., Wei Shi, Chao Li*, Zihu Zhang, Meng Cheng, Haodong Gu, Dalton S. Hardisty, Sean J. Loyd, Malcolm W. Wallace, Ashleigh vS. Hood, Kelsey Lamothe, Benjamin J. W. Mills, Simon W. Poulton, Timothy W. Lyons. Uncovering the Ediacaran phosphorus cycle. *Nature*, 2023, in press, doi: <https://doi.org/10.1038/s41586-023-06077-6>.
- Derry, L.A., 2010. A burial diagenesis origin for the Ediacaran Shuram–Wonoka carbon isotope anomaly. *Earth Planet. Sci. Lett.* 294, 152–162.
- Evans, D.A.D., 2006. Proterozoic low orbital obliquity and axial-dipolar geomagnetic field from evaporite palaeolatitudes. *Nature* 444, 51–55.
- Fike, D.A., Grotzinger, J.P., Pratt, L.M., Summons, R.E., 2006. Oxidation of the Ediacaran Ocean. *Nature* 444, 744–747.
- Gingras, M., Hagadorn, J.W., Seilacher, A., Lalonde, S. V., Pecoits, E., Petrash, D., Konhauser, K.O., 2011. Possible evolution of mobile animals in association with microbial mats. *Nat. Geosci.* 4, 372–375.
- Grotzinger, J.P., Fike, D.A., Fischer, W.W., 2011. Enigmatic origin of the largest-known carbon isotope excursion in Earth’s history. *Nat. Geosci.* 4, 285–292.
- Husson, J.M., Maloof, A.C., Schoene, B., Chen, C.Y., Higgins, J.A., 2015. Stratigraphic expression of earth’s deepest $\delta^{13}\text{C}$ excursion in the wonoka formation of South Australia. *Am. J. Sci.* 315, 1–45.
- Kaufman, A.J., Corsetti, F.A., Varni, M.A., 2007. The effect of rising atmospheric oxygen on carbon and sulfur isotope anomalies in the Neoproterozoic Johnnie Formation, Death Valley, USA. *Chem. Geol.* 237, 47–63.
- Knauth, L.P., Kennedy, M.J., 2009. The late Precambrian greening of the Earth. *Nature* 460, 728–732.
- Lu, M., Zhu, M., Zhang, J., Shields-Zhou, G., Li, G., Zhao, F., Zhao, X., Zhao, M., 2013. The DOUNCE event at the top of the Ediacaran Doushantuo Formation,

- South China: Broad stratigraphic occurrence and non-diagenetic origin. *Precambrian Res.* 225, 86–109.
- Luz, B., Barkan, E., 2010. Variations of $17\text{O}/16\text{O}$ and $18\text{O}/16\text{O}$ in meteoric waters. *Geochim. Cosmochim. Acta* 74, 6276–6286.
- McFadden, K.A., Huang, J., Chu, X., Jiang, G., Kaufman, A.J., Zhou, C., Yuan, X., Xiao, S., 2008. Pulsed oxidation and biological evolution in the Ediacaran Doushantuo Formation. *Proc. Natl. Acad. Sci.* 105, 3197–3202.
- Neretin, L.N., Volkov, I.I., Böttcher, M.E., Grinenko, V.A., 2001. A sulfur budget for the Black Sea anoxic zone. *Deep Sea Res. Part I Oceanogr. Res. Pap.* 48, 2569–2593.
- Oba, Y., Poulson, S.R., 2009. Oxygen isotope fractionation of dissolved oxygen during abiological reduction by aqueous sulfide. *Chem. Geol.* 268, 226–232.
- Ren, R., Guan, S.W., Zhang, S.C., Wu, L., Zhang, H.Y., 2020. How did the peripheral subduction drive the Rodinia breakup: Constraints from the Neoproterozoic tectonic process in the northern Tarim Craton. *Precambrian Res.* 339, 105612.
- Shi, W., Li, C., Luo, G., Huang, J., Algeo, T.J., Jin, C., Zhang, Z., Cheng, M., 2018. Sulfur isotope evidence for transient marine-shelf oxidation during the Ediacaran Shuram Excursion. *Geology* 46, 267–270.
- Shields, G.A., Mills, B.J.W., Zhu, M., Raub, T.D., Daines, S.J., Lenton, T.M., 2019. sustained by coupled evaporite dissolution and pyrite burial. *Nat. Geosci.* 12.
- Siang, H.Y., Tahir, N.M., Malek, A., Isa, M.A.M., 2017. BREAKDOWN OF HYDROGEN SULFIDE IN SEAWATER UNDER DIFFERENT RATIO OF DISSOLVED OXYGEN / HYDROGEN SULFIDE. *Malaysian J. Anal. Sci.* 21, 1016–1027.
- Tahata, M., Ueno, Y., Ishikawa, T., Sawaki, Y., Murakami, K., Han, J., Shu, D., Li, Y., Guo, J., Yoshida, N., Komiya, T., 2013. Carbon and oxygen isotope chemostratigraphies of the Yangtze platform, South China: Decoding temperature and environmental changes through the Ediacaran. *Gondwana Res.* 23, 333–353.
- Wang, R., Shen, B., Lang, X., Wen, B., Mitchell, R.N., Ma, H., Yin, Z., Peng, Y., Liu, Y., Zhou, C., 2023. A Great late Ediacaran ice age. *Natl. Sci. Rev.* 12, 4683–4698.
- Wang, Y., Chen, D., Liu, M., Liu, K., Tang, P., 2022. Ediacaran carbon cycling and Shuram excursion recorded in the Tarim. *Precambrian Res.* 377, 106694.
- Wu, S., Jeschke, C., Dong, R., Paschke, H., Kusch, P., Knöller, K., 2011. Sulfur

transformations in pilot-scale constructed wetland treating high sulfate-containing contaminated groundwater: A stable isotope assessment. *Water Res.* 45, 6688–6698.

Xiao, S., Bao, H., Wang, H., Kaufman, A.J., Zhou, C., Li, G., Yuan, X., Ling, H., 2004. The Neoproterozoic Quruqtagh Group in eastern Chinese Tianshan: Evidence for a post-Marinoan glaciation. *Precambrian Res.* 130, 1–26.

Xu, B., XIAO, S., ZOU, H., CHEN, Y., LI, Z., SONG, B., LIU, D., ZHOU, C., YUAN, X., 2009. SHRIMP zircon U–Pb age constraints on Neoproterozoic Quruqtagh diamictites in NW China. *Precambrian Res.* 168, 247–258.

Xu, Y., Schoonen, M.A., Nordstrom, D., Cunningham, K., Ball, J., 1998. Sulfur geochemistry of hydrothermal waters in Yellowstone National Park: I. the origin of thiosulfate in hot spring waters. *Geochim. Cosmochim. Acta* 62, 3729–3743.

Reviewer #1 (Remarks to the Author):

Review of Wang et al., Nat Comms 2023

This is my second review of Wang et al., 's paper submitted to Nature Communications. I believe the authors have sufficiently rebutted, acknowledged, and incorporated my comments and concerns as well as those of other reviewers. I do note that it remains to be observed in natural modern environments whether H₂S oxidation in natural water bodies incorporates the predicted $\Delta^{17}\text{O}$ signal from O₂. I believe the authors should acknowledge this unknown in their final version of the manuscript. I have attached a pdf with some suggested grammatical improvements.

Again, I believe this paper will have a large impact on the community and motivate many more CAS-based $\Delta^{17}\text{O}$ studies. I believe that this will encourage a step change in the resolution of $\Delta^{17}\text{O}$ sulfate records which may provide many new important insights for the community. Therefore I am supportive of this work for publication and recommend that it be accepted.

In their review of the second version of this manuscript, reviewer #1 added some comments to the manuscript file. These comments were forwarded to the authors, who replied as included in this Peer Review File.

Reviewer #2 (Remarks to the Author):

The revised manuscript was improved by the 3 reviewer comments. Authors thoughtfully responded and incorporated comments as appropriate. The manuscript is compelling and of wide interest to readers. There is one minor comment below otherwise this manuscript should be accepted.

Line 193-194: "the reverse may not necessarily be true." What is the reverse? Early diagenesis, negative correlation, non-correlation? Please be more specific in this sentence.

Reviewer #3 (Remarks to the Author):

The authors did an impressive job in revising this manuscript. All my concerns raised in my last round of reviews have been well addressed. The revised version looks much better, in both writing and logic flow. I appreciate the authors' efforts in improving this manuscript. With great pleasure, now I recommend the editor accept this manuscript for publication in Nature Communications.

Response to Reviewers' Comments

Reviewer #1 (Remarks to the Author):

Review of Wang et al., Nat Comms 2023

This is my second review of Wang et al., 's paper submitted to Nature Communications. I believe the authors have sufficiently rebutted, acknowledged, and incorporated my comments and concerns as well as those of other reviewers. I do note that it remains to be observed in natural modern environments whether H₂S oxidation in natural water bodies incorporates the predicted $\Delta^{17}\text{O}$ signal from O₂. I believe the authors should acknowledge this unknown in their final version of the manuscript. I have attached a pdf with some suggested grammatical improvements.

Again, I believe this paper will have a large impact on the community and motivate many more CAS-based $\Delta^{17}\text{O}$ studies. I believe that this will encourage a step change in the resolution of $\Delta^{17}\text{O}$ sulfate records which may provide many new important insights for the community. Therefore, I am supportive of this work for publication and recommend that it be accepted.

Response: We appreciate the reviewer's positive evaluation and the valuable grammatical improvements made (all have been revised). We acknowledge the reviewer's concern regarding the incorporation of the predicted $\Delta^{17}\text{O}$ signal from O₂ during H₂S oxidation in natural water bodies. In response, we have now added in lines 318-320, "*Additionally, further research efforts are required to validate if sulfate derived from H₂S oxidation in natural water bodies indeed incorporates the $\Delta^{17}\text{O}$ signal of atmospheric O₂*".

Reviewer #2 (Remarks to the Author):

The revised manuscript was improved by the 3 reviewer comments. Authors thoughtfully responded and incorporated comments as appropriate. The manuscript is compelling and of wide interest to readers. There is one minor comment below otherwise this manuscript should be accepted.

Line 193-194: "the reverse may not necessarily be true." What is the reverse? Early diagenesis, negative correlation, non-correlation? Please be more specific in this sentence.

Response: Thanks. We could have expressed it better. This sentence has been revised to read as "*While late meteoric diagenesis can generate a positive correlation between $\delta^{13}C_{carb}$ and $\delta^{18}O_{carb}$, the existence of this correlation does not necessarily confirm meteoric diagenesis.*" at lines 194-196.

Reviewer #3 (Remarks to the Author):

The authors did an impressive job in revising this manuscript. All my concerns raised in my last round of reviews have been well addressed. The revised version looks much better, in both writing and logic flow. I appreciate the authors' efforts in improving this manuscript. With great pleasure, now I recommend the editor accept this manuscript for publication in Nature Communications.

Response: We once again extend our appreciation for the reviewer's time and effort.